# The influence of proximity and knowledge base on recombination innovation in R&D collaboration

Ding Nan 🔘 *

School of Economics and Management, University of Science and Technology Beijing, Beijing, China

* nanding@ustb.edu.cn

## Abstract

Recombination innovation invented during organizations' research and development (R&D) collaborations is a vital mechanism for creating new technological knowledge for organizations. This study aims to reveal the contribution mechanism of different dimensions of proximity to the recombination innovation at the collaborative dyad level and how this mechanism depends on the characteristics of organizations' knowledge base structuration. We conceptualize that the interdependence among knowledge elements in the knowledge base forms the knowledge space of the organization and build a theoretical framework to explain the interactive effect of proximity and organizations' knowledge base characteristics on collaborative recombination innovation. We validated the theoretical hypotheses using Logit regression models based on the longitudinal data of 150 organizations in the global nanotechnology industry. As demonstrated by our findings, technological proximity exerts a negative effect, while geographic proximity exerts an inverted U-shaped effect on collaborative organizations' joint recombination innovation. Organizations' knowledge base decomposability plays a negative role in moderating the effect of technological proximity and plays a positive role in regulating the effect of geographic proximity. In contrast, the degree centrality of the knowledge elements positively moderates the effect of both technological and geographic proximity.

## Introduction

Recombination innovation is a process of creatively combining technological components or re-configuring existing knowledge components resulting in the development of new inventions [1, 2]. In the last years, recombination innovation has gained increasing importance since the miniaturization and architectural changes of complex products have enhanced the relevance of combining previously unrelated knowledge components into one product. Studies also show that the atypical combination of conventional knowledge spanning a broader range of technological domains produces inventions with high impacts [3]. Notably, existing studies have highlighted the importance of organizations' external R&D collaboration as a critical mechanism to recombine novel knowledge from partners, the process of which would spur

**Data Availability Statement:** The data is deposited at the link: https://dataverse.harvard.edu/dataset.xhtml?persistentId=doi:10.7910/DVN/AEGKNE To cite: Nan, Ding, 2023, "The influence of proximity and knowledge base on recombination innovation in R&D Collaboration", https://doi.org/10.7910/

DVN/AEGKNE, Harvard Dataverse, V1, UNF:6:3MTvUhAlLoVBGYkkBmwjLg== [fileUNF].

**Funding:** The author(s) received no specific funding for this work.

**Competing interests:** The authors have declared that no competing interests exist.

more recombination innovation [4–6]. To a certain extent, recombination innovation tremendously rests with the inter-organizational collaborative process as it provides channels of integrating heterogeneous knowledge elements, practice, and mental models. As a form of collaboration that crosses organizational boundaries, inter-organizational collaboration can help meet the heterogeneous demand for innovation resources.

However, there are a large number of cases of dissolution and failure of inter-organizational collaborations. The high failure rate of inter-organizational collaborations restricts the development of organizations' recombination innovation. To date, our comprehension of the relationship between partner selection and organization collaborative recombination innovation is scattered and lacks a systematic research perspective. Therefore, this paper attempts to explore the characteristics of collaborative partners that influence organizations' collaborative recombination innovation based on the proximity perspective, using technological and geographical proximity as well as the underlying knowledge base characteristics.

Collaboration between organizations is often achieved in a cross-domain manner or across geographical distances. Studies have confirmed that technological and geographic proximity among collaborating agents influences the recombination innovation at the collaborative dyad level [7–9]. Technological proximity usually determines the extent of understanding of partners' knowledge. Geographical proximity influences the ease of face-to-face communication between the partners, determining the knowledge transmission efficiency [10, 11]. In addition, at different levels of technological and geographic proximity, organizations encounter knowledge components originating from different technological and geographic origins [10, 12]. Therefore, technological and geographical proximity among collaborating partners may influence the collaborative recombination innovation. In fact, organizations obtain knowledge from technological and geographically distant partners, which helps them to deviate from the familiar "thinking model" and spot new recombination opportunities. However, despite the recognition of the proximity effect on collaborative innovation, current research fails to throw light upon two essential aspects.

First, in the existing literature, proximity and collaborative innovation are directly linked in theorization, while the underlying transmission mechanism is, to a large extent, black-boxed [13, 14]. In the economic geography literature, studies have investigated various proximity dimensions, and their effects on the formation and maintenance of collaborations in the context of inter-regional [15] and inventor collaboration [16]. In addition, the effect of proximity on the knowledge sharing and production in inter-organizational alliance is well documented [10]. While in the theorization of the effects of proximity during organization-to-organization collaborations, the role of underlying organizational characteristics remains less salient. Research has rarely examined how proximity and organizational characteristics jointly spur innovation at the collaborative dyad level. Second, existing literature highlights that studies should focus on knowledge space within organizational boundaries when examining inter-organizational collaboration [17]. While, examining the current economic geography literature, few studies have applied the concept of "knowledge space" at the organizational level and link it to the outcomes of cooperation.

In order to fill these gaps, we focus on organization-to-organization collaboration to further contribute to strengthening this body of work on the collaborative nature of innovation and the role of proximity in this process. This paper focuses on collaborating organizations' knowledge base characteristics, reflecting the conception that the organizations combine knowledge elements of different domains during innovative process, and the knowledge elements are coupled together to form the organization's knowledge base. Knowledge base characteristic represents the manner of the interdependence among knowledge elements and it lays the foundation for an organization's ability to recombine partners' knowledge. Previous studies

put forward that a knowledge base is a network of knowledge elements in essence [18–21] that have great implications for the formation of collaborations [17, 22], collaborative effectiveness [22, 23], and learning of partners' knowledge [24]. The interdependence among knowledge elements can be conceptualized as "knowledge space" within the organization and it represents the pattern of how knowledge elements are coupled with each other. Knowledge space influences an organization's knowledge searching process [25]. Notably, knowledge space remarkably facilitates knowledge integration during inter-organizational collaborations [19, 23, 26]. Based on this, it is important to explore the role of knowledge base characteristics in studying the relationship between technological proximity, geographical proximity and collaborative recombination innovation.

Overall, this paper probes into the influence of technological and geographical proximity on collaborative recombination innovation based on the proximity research perspective and the knowledge-based view. And this paper introduces knowledge base characteristics as a moderating mechanism to further open the "black box" of the relation between multidimensional proximity and recombination innovation. We mainly concentrate on two marked aspects of the knowledge base characteristics, which are knowledge base decomposability, i.e., the extent to which knowledge elements are connected with each other forming clustered structures in the knowledge space, and degree centrality of knowledge elements, i.e., the extent to which knowledge elements are connected to other knowledge elements within and beyond the organizational boundary. These two dimensions of knowledge base characteristic of an organization can represent two different learning strategies. Knowledge base decomposability usually originates from an organization's expertise in specialized fields of knowledge [18, 21]. These organizations often rely on their deep utilization of domain knowledge to maintain existing advantages. The degree centrality of knowledge elements in the knowledge space, on the other hand, usually originates from the broader connectedness of technologies within the organization [21]. Organizations tend to rely more on the combinatorial potential of knowledge as well as external knowledge acquisition to exploit the internal technologies fully. To empirically test the hypotheses, we utilized the data of 350 R&D collaborative dyads comprising 150 organizations in the global nanotechnology industry for 18 years (2000–2018).

In this paper, we advance the recombination innovation literature by analyzing how recombination innovation is jointly developed by collaborating organizations through the lens of multidimensional proximity. By effectively identifying collaborating partners' characteristics through the proximity dimension, it helps organizations to more effectively access heterogeneous knowledge pools and spot recombination opportunities. Also, we go beyond the traditional conceptualization of the knowledge base and focus on the interdependence among knowledge elements in the knowledge space to further shed light on how recombination innovation is jointly produced. We analyze how proximity and knowledge base may interact and spur more recombination innovation output at the collaborative dyad level.

## Theory and hypotheses

### Proximity, knowledge base and recombination innovation

R&D collaborations are especially important in the process of recombination innovation as the exposure to new knowledge would offset organizations' weakness of limited knowledge variety [27, 28]. Moreover, engaging with partners can drive organizations to deviate from familiar search elements, giving birth to a more comprehensive perspective on the technological landscape as well as greater flexibility in problem-solving. Thus, organizations can uncover valuable combinations of knowledge elements raising the opportunity to reach high peaks on the technological landscape [29]. In this sense, joint recombination innovation with partners

not only involves the expansion of the organization's knowledge base but also drives to reconfigure and make use of their existing knowledge elements creatively [30].

In reality, organizations create social relationships with partners along different dimensions of proximity [31–33]. Boschma [7] puts forward the analytical framework of five dimensions of proximity which are cognitive proximity, organizational proximity, social proximity, institutional proximity, and geographical proximity. With the notion of cognitive proximity or technological proximity, actors share the same knowledge base and thus are able to identify, interpret, and exploit each other's knowledge. Organizational proximity is the extent to which relations are shared in the same organizational arrangement. Actors in the same organization usually enjoy a higher level of organizational proximity. On the other hand, social proximity indicates the extent to which actors have social relations and are embedded in the same social context. Actors with institutional proximity often share the same set of habits, routines, established practices, rules, norms and values. It includes both formal (such as laws and rules) and informal institutions (such as cultural norms and habits). Geographical proximity refers to the spatial or physical distance between actors [7].

On account of Boschma's theory, different dimensions of proximity limit the extent to which organizations benefit from knowledge co-cocreation with partners [7, 34]. Essentially, proximity promotes or hinders knowledge spillover in collaborations following several mechanisms. On the one hand, proximity gives rise to the common knowledge base, language, and context of mutual learning among collaborating agents [30, 35, 36]. On the other hand, at a lower level of proximity, organizations encounter the knowledge from distinct technical fields, disciplines, regions and with a higher level of novelty.

In addition, organizations' knowledge base forms the internal knowledge space and lays the foundation of adaptation, association, and coupling of the organizations' knowledge elements, which exerts influence on the final emergence of collaborative recombination innovation with partners. In the viewpoint of Knowledge based view (KBV), organizations gradually develop different types of knowledge [37, 38] as well as the knowledge about the interdependence among knowledge components [18]. In particular, in an organization's knowledge space, coupled knowledge elements are often considered together by organizations as they are perceived as closely related technologies [25, 39]. Accordingly, an organization's knowledge about the interdependence of knowledge elements predominantly acts as a "cognitive map" or principal theory that guides the recombination of external knowledge. On account of this mechanism, the internal knowledge space of an organization reflects what type of knowledge can be recombined into organizations' knowledge base and how much is the recombinant potential of the technologies within the organizations.

To sum up, recombinant search during external collaborations proceeds through an evolutionary process: organizations sample from the pool of technological possibilities and adjust the direction of search as evidence is generated and inferences derived [29, 40, 41]. Specifically, knowledge along different dimensions of proximity helps to provide novel knowledge, complementary resources, and technological experiences that are needed to spur joint recombination innovation. In the meanwhile, knowledge space drives organizations to "discard" those knowledge elements that do not match existing knowledge elements and thus guides the joint innovation process.

## Knowledge base characteristic: The knowledge base decomposability and the degree centrality of knowledge elements

In this paper, we focus on two salient characteristics of knowledge base: knowledge base decomposability and degree centrality of knowledge elements.

The concept of decomposability comes from the analysis of product systems [42, 43]. Decomposability refers to the degree to which a system's components can be separated and recombined into new configurations with little loss of functionality [42]. Decomposability reflects the cohesiveness of the knowledge space [18, 19, 39] in that when the knowledge base has a high degree of decomposability, knowledge elements are coupled with each other forming many triad structures or clusters. Here, knowledge elements are relatively independent of one another because the ties within clusters are strong, while they are weak across clusters [42–44]. By contrast, when the degree of decomposability is low, there are pervasive interdependence among knowledge elements, while the knowledge space does not show clear triads or clusters.

Differently, the degree centrality of knowledge elements indicates the level of connectedness of knowledge elements. It reflects the combinatorial potential of knowledge components. When the degree centrality of knowledge elements is high, there is a natural relatedness among technologies held by the organization. Also, researchers in the organization have beliefs and desirability about combining internal technologies and have fully explored their combinatorial opportunities [19]. Accordingly, researchers have accumulated experience in combining internal technologies and have utilized them in many products. By contrast, when the degree centrality of knowledge elements is low, researchers have a low level of belief about the scientific, technological, or commercial value of organizations' knowledge elements [45]. Consequently, the combinatorial potential of an organization's technologies is low.

We argue that the influence of different dimensions of proximity on joint recombination innovation is conditioned on knowledge base characteristics. Thus, we first put forward the baseline hypotheses on the relationship between proximity and recombinant innovation in the following part.

## Technological proximity and recombination innovation

We argue that technological proximity inhibits recombination innovation in R&D collaboration due to several reasons. Recombination innovation is a process of combining knowledge elements within and across technological domains [20]. In this sense, it is, to a large extent, influenced by the knowledge content variety and novelty that can be considered [9, 46]. Too many similarities in the knowledge base would hurt the alternative knowledge that can be considered limiting the recombination scope [47]. Thus, when collaborating partners face technologies that are similar to their own knowledge, it would confine them within limited recombination space and inhibit them from discovering valuable knowledge combinations.

By contrast, as technological proximity decreases, organizations get access to heterogeneous and diverse knowledge elements. To put it in another way, the technological distance between partners leads to a higher exposure to heterogeneous knowledge and a higher probability of discovering novel combinations of knowledge. Moreover, in the viewpoint of previous studies, the complementariness between knowledge elements is raised because complementariness often comes from related but different domains of expertise [25]. Lastly, previous studies show that a lower level of recombination generally carries low uncertainty but may be short of novelty [45]. Differently, an increased technological distance between partners is conductive to inventions with a high level of recombination and as a result, carries a high degree of novelty [48]. In accordance with the above elaboration, we put forward the baseline hypothesis:

**Hypothesis 1**: *Technological proximity between collaborating partners has a negative effect on the recombination innovation in R&D collaboration.*

## Geographic proximity and recombination innovation

Geographic proximity reflects a different dimension of relational characteristic among partnering organizations compared to technological proximity. Technological proximity represents the overlap of technological background and expertise of the organizations, while geographic proximity is the spatial distance between the collaborating organizations. Spatial propinquity encourages frequent face-to-face interactions, thus stimulating the emergence of interpersonal networks across the boundaries of partner organization. This, in turn, increases the opportunities for face-to-face communication of the inventors, facilitating the development of relational trust. Therefore, partnering organizations develop, learn, and adjust over time the idiosyncratic languages needed for the sharing of 'fine-grained information' and tacit knowledge. Although organizations located in the closed vicinity of each other draw on the same set of regional knowledge and tend to share similar values, norms, and "thinking models", it does not necessarily mean that these organizations have a larger technological proximity. Therefore, geographical proximity and technological proximity are independent from each other and are both essential to determine recombination innovation at the collaborative dyad level.

As the geographic proximity of collaborative partners increases, it elevates recombination innovation in R&D collaborations following several mechanisms. First, the recombination innovation process often requires tacit knowledge about reconfiguring and reutilizing existing technological components in new context [49]. In a situation where geographic proximity is large and collaborative agents are located close to each other, the probability that researchers meet and discuss face-to-face frequently is high. On account of this reason, there is more chance to transfer valuable tacit knowledge that is conducive to recombination innovation. Moreover, recombinant innovation always entails a long experimentation process, including trial and error. Hence, organizations should not only know about their own technologies but also acquire and understand partners' key technological components. In reality, collaborating agents located close to each other usually have similar values, norms, and cultures, which gives rise to communication efficiency as well as the motivation of organizations to learn from each other. Accordingly, an ameliorated learning efficiency and motivation would spur more joint recombination innovation. Second, geographic proximity often improves monitoring and managerial ability to lay relationship building processes. Being spatially close to collaborating partners always means a high level of relationship quality, promotes mutual understanding, and supports creativity and information exchange.

However, as geographic proximity increases and exceeds a certain threshold, it becomes detrimental to recombination innovation. The reason lies in that when collaborating agents are located too close, similar technical backgrounds and capabilities result in the draining of knowledge variety and standardization of know-how [50]. Aside from that, these factors also prohibit organizations from initiating new projects and exploring new technology fields. Consequently, geographic proximity reduces the possibility for novel recombination of different skills [51]. Finally, existing literature show that knowledge spillovers tend to be spatially bounded due to the inherent tacit nature of much of the core knowledge [34, 52]. Actually, locating too close to each other may lead to unwanted knowledge leakage, which has detrimental effects on organizations' innovation process. In accordance with the above elaboration, we put forward the baseline hypothesis as:

**Hypothesis 2**: *Geographical proximity of partners has an inverted U-shaped effect on the recombination innovation in R&D collaboration.*

## Moderating effect of knowledge base characteristic

**The moderating role of the degree of decomposability of the knowledge base.** We argue that knowledge base decomposability negatively moderates the relation between technological proximity and recombination innovation in R&D collaboration. Notably, a knowledge base with a higher level of decomposability can be split into distinct clusters of technologies and know-how, thus when recombining internal knowledge with partners' new knowledge, organizations do not have to reconfigure the entire knowledge base. As a result, the recombination of new knowledge can be conducted in certain knowledge clusters, significantly reducing the threshold of changing the knowledge base configuration and allowing for enhanced exploration of new knowledge [18]. Consequently, collaborating organizations can achieve a higher level of recombination innovation when the technological proximity is small and the degree of decomposability of the knowledge base is high. Furthermore, when knowledge base decomposability is high, collaborating partners have extensive experience in combining knowledge elements in certain technological domains [53] and consider closely related knowledge elements as a unified set of technology. As a result, the structured and clustered knowledge base drives the search cost significantly lower compared to irregular structured diverse knowledge. On account of this mechanism, it leads to more efficient recombination of heterogenous knowledge in a situation of smaller technological proximity.

However, at large technological proximity, a higher degree of knowledge base decomposability does not show obvious superiority of recombining familiar knowledge. Altogether, the slope of the negative linear relation between technological proximity and joint recombination innovation becomes larger when the degree of knowledge base decomposability increases. In accordance with the above elaboration, we put forward hypothesis as:

**Hypothesis 3**: *Organization's knowledge base decomposability negatively moderates the relation between technological proximity and recombination innovation in R&D collaboration.*

We also argue that knowledge base decomposability escalates the positive effect of heterogenous knowledge at smaller geographic proximity and mitigates the negative effect of knowledge homogeneity at larger geographic proximity. Nonetheless, technological expertise is often dispersed into specialized technological territories for organizations with a high level of knowledge base decomposability. In such cases, knowledge base adaptation is achieved through changes in certain domains, and organizations do not have to change the entire knowledge space when recombining new knowledge. In accordance with this mechanism, organizations can explore more paths into new regions and are better able to discover new recombinant opportunities offered by collaborating partners [22]. Hence, the positive effect of heterogenous knowledge resulting from smaller geographic proximity is elevated. Furthermore, organizations' cognition about knowledge interdependence in specialized technological territories helps to solve complex problems that are often encountered when collaborating with distant partners. As such, organizations' relative absorptive capacity is elevated. Thus, we would expect that knowledge base decomposability would pivot the slope of novelty accessed through smaller geographic proximity upward.

Moreover, organizations having a knowledge base with a high level of decomposability often possess knowledge about how to link and combine existing knowledge elements from specialized technological domains. Therefore, when collaborating with local partners, organizations can rely on their experience and come up with more valuable recombination of existing knowledge elements. Also, having a knowledge base with a high level of decomposability, organizations can mitigate the danger of knowledge leakage to closely located partners [53]. As indicated in the previous studies, a clustered knowledge base is difficult to replicate by others

[21] because organizations' expertise in specialized technological domains consists of tacit knowledge related to interrelations among knowledge components. Moreover, this expertise tends to reside in informal communication channels among different researchers, and R&D units, which makes it even harder to imitate [17, 54, 55]. Hence, the degree of knowledge base decomposability would mitigate the negative effect caused by geographic proximity.

Taken together, we expect the flattening of inverted U-shaped relation leading to a positive moderation effect [56]. Based on above arguments, we put forward hypothesis as:

**Hypothesis 4**: *Organization's knowledge base decomposability positively moderates the relation between geographic proximity and recombination innovation in R&D collaboration.*

**The moderating role of degree centrality of knowledge elements.** We argue that the degree centrality of knowledge elements in the knowledge space of the collaborating agents would positively moderate the relation between technological proximity and recombination innovation in R&D collaboration. Larger degree centrality of knowledge elements means that knowledge elements have a clear signal of legitimacy [45], which motivates researchers to explore more recombination opportunities [18]. Thus, more research projects are initiated as resources and attention are always directed towards popular research topics. In accordance with this mechanism, through repeatedly creating unique knowledge ties, tasks and interactions are routinized within the organization [57]. In this situation, when technological proximity is small, researchers' beliefs in combining internal knowledge reduce their motivation to learn new knowledge of their partners [24]. Indeed, researchers are usually positively biased towards the value of their own knowledge and are negatively biased towards external new knowledge [58]. These biases become even more pronounced when the combinatory potential of their own knowledge is higher, i.e., the degree centrality of knowledge elements in the knowledge space is high. Hence, organizations perceive new technologies as unnecessarily risky and costly, which reduces their motivation to recombine technologically distant knowledge. As such, we believe that the degree centrality of knowledge elements would reduce the strength of positive mechanisms associated with smaller technological proximity.

Conversely, when the technological proximity is large, organizations encounter familiar technologies during R&D collaborations. Previous studies put forth that when organizations have a history of recombining components in unique ways, they have valuable experience in understanding and dealing with idiosyncratic ties among existing components. As such, when the degree centrality of organizations' knowledge elements is high, researchers can find unique linkages of similar knowledge components, the process of which would elevate the level of recombination innovation that can be achieved by collaborating agents with larger technological proximity.

Taken together, we expect the slope of the negative linear relation becomes smaller, leading to a positive moderation effect. On the basis of the above exploration, we put forward hypothesis as:

**Hypothesis 5**: *Organizations' degree centrality of knowledge elements positively moderates the relation between technological proximity and recombination innovation in R&D collaboration.*

We argue that the degree centrality of knowledge elements elevates the positive effect of heterogenous knowledge at smaller geographic proximity and mitigates the negative effect of knowledge homogeneity at larger geographic proximity. As the degree centrality of knowledge elements increases, knowledge elements are frequently used in many products facilitating knowledge reuse in organizations. As suggested in the previous studies, knowledge reuse

propels researchers' capacity to exchange ideas and engage in collective learning, thereby improving the organization's absorptive capacity [35]. Accordingly, organizations' capability to value, assimilate, and learn partner technologies across a larger geographic distance is ameliorated. Aside from that, knowledge element centrality boosts organizations' capability to find complementary domains of expertise owing to the reason that these knowledge elements have been widely utilized in many contexts. Hence, organizations can reap more benefits from collaborating with geographically distant partners, given that their partners' knowledge is more successfully learned and integrated, and also, more complementariness between technologies can be spotted. In sum, we would expect that the degree centrality of knowledge elements would pivot the slope of novelty accessed through larger geographic proximity upward.

Alternatively, local collaborations among geographically close agents usually have similar values, languages, and "thinking models". f In this situation, when the degree centrality of knowledge elements is high, organizations' combinatory experience of internal knowledge would elevate the marginal benefit of recombining similar and cognitive elements. Consequently, the degree centrality of knowledge elements would mitigate the knowledge homogeneity effect caused by larger geographic proximity.

Taken together, we would expect the flattening of inverted U-shaped relation leading to a positive moderation effect [56]. On the basis of the above exploration, we put forward the hypothesis as:

**Hypothesis 6**: *Organizations' degree centrality of knowledge elements positively moderates the relation between geographic proximity and recombination innovation in R&D collaboration.*

According to the above relevant literature and hypotheses, we present the following framework (Fig 1).

## Methods

### Data and sample

To test our hypotheses, we restrict our analysis to the global nanotechnology industry. The development of the nanotechnology industry is driven by collaborations among different organizations, the process of which requires specialized knowledge from diverse sectors dispersed around the globe [59]. Thus, the nanotechnology sector is a suitable arena to study R&D collaboration and recombination innovation. We utilized nanotechnology patent data filed by USPTO (United State Patent and Trademark Office) to identify organizations (including firms, universities, and research institutions) in the nanotechnology industry. Specifically, we downloaded all the nanotechnology-related patents using the search phrase "977", filed between 1978 and 2018 from the USPTO. In USPTO patent dataset, all the nanotech-related patents contain the phrase of "977". Then, as our analysis is conducted on organizational level and we focus on organizations and their granted patents, patents filed to the individual inventors are excluded from the dataset. Our initial set of data consists of a total of 11000 patents.

In this study, we rely on joint patents to identify R&D collaborations among organizations, as well as to measure recombination innovation performance. The joint patent is assigned to or jointly owned by more than one organization [60]. Due to this measurement choice and procedure, only organizations that have at least one joint patent application are considered in our sample. Also, in the purpose of incorporating the organization's patenting history into account, only organizations with at least one patent application before the observation year are included in our sample. It is because several measurements in our analysis require the organization's patenting history. For example, we rely on organizations' previous patents to obtain their technology portfolio and calculate technological proximity. We also rely on

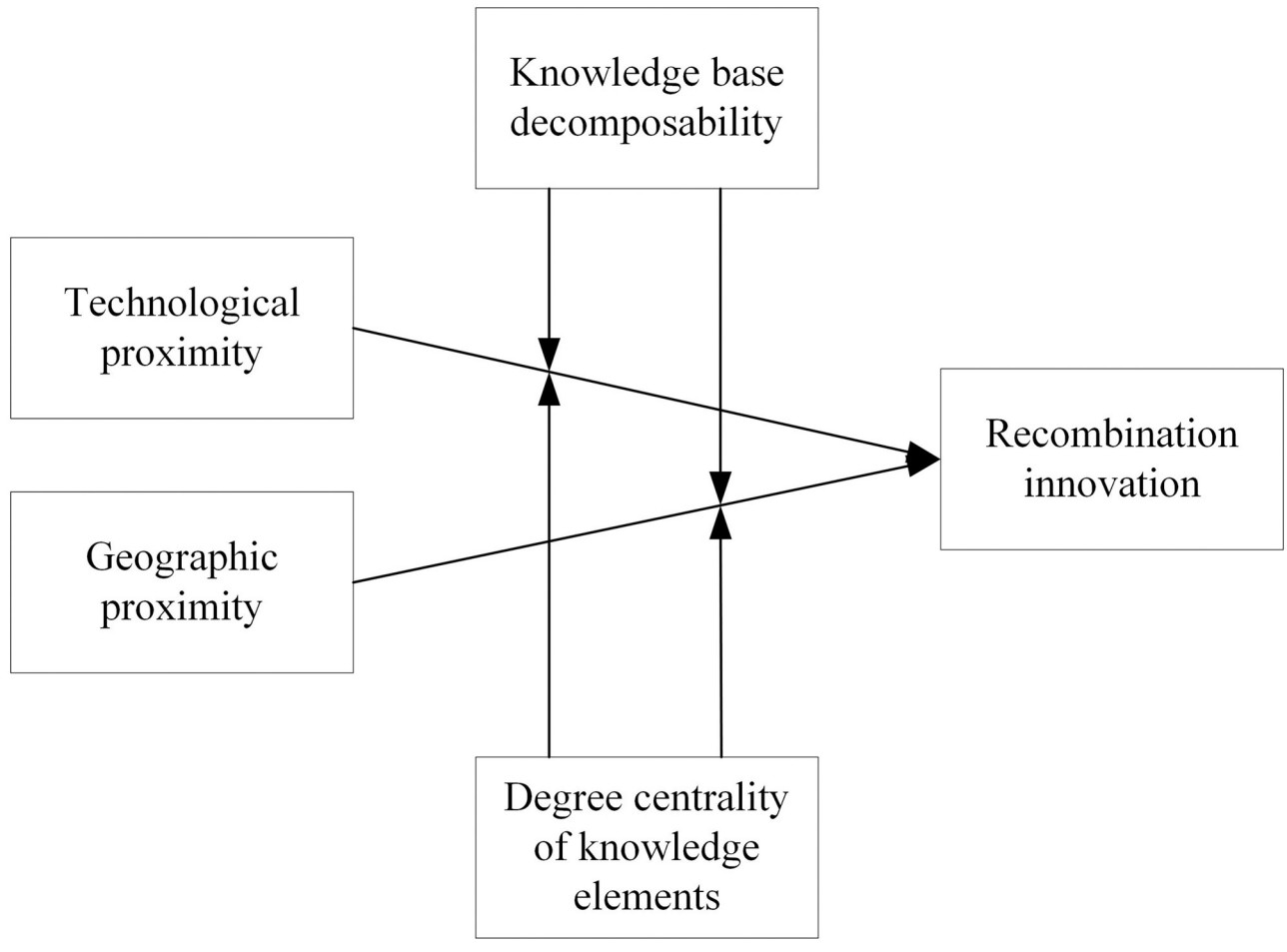

**Fig 1. Analytical framework.**

organizations' previous patents to calculate accumulated patent counts and obtain the measure of past performance. We started our analysis from the year 2000 as from this year on, nanotechnology patents increased dramatically. We ended our analysis in 2018 as our patent data is only updated till 2018. Finally, this procedure leads to a dataset comprising 150 organizations in nanotechnology in the period of 2000 to 2018, forming 350 organization-year observations. Our study is conducted through collaborative-dyad level (or joint patent level) analysis.

## Measures

**Dependent variables.** In line with previous studies, we define recombination innovation *(Recom)* as novel combinations of International Patent Classification (IPC) that have not appeared on any of the collaborating partners' previous patents [20, 45, 59, 61]. When granting a patent, the patent office usually assigns several technological domains that the patent is associated to. Each technological domain is represented by a unique IPC class. If two patents contain the same IPC, it means that these two patents belong to a same technological domain. A combination of two different IPCs means using two technologies within the same invention. In this paper, recombination innovation is measured at the joint patent level, with the indicator equal to 1 if the joint patent contains a new combination of IPC and 0 otherwise. A

combination of IPC classes is considered new if no patents of the two organizations in the previous years had the same combination of IPC classes. For example, organization A possesses IPC class X and Z and organization B possesses IPC class Y and K. If organization A and B collaborate and create patent 1 with IPC class X, Y and patent 2 with IPC class M, then these two patents are all counted as recombination innovation jointly developed by A and B. It is because patent 1 contains the combination of X and Y that has not appeared in the previous patents of both organization A and B. Patent 2 introduces a new technology M for both organizations that they have not used or combined previously on their own. In this way, our measurement of recombination innovation includes co-patents that contain a new IPC classification that has not appeared in the previous patents of collaborating organizations, and the new combinations of existing IPCs. To guarantee the use of solely successful patent applications, only granted patents instead of applied patents are considered. However, by using the date of the application instead of the date of the grant, we reduce distortion because the time horizon of the grant procedure essentially depends on the characteristics of the patent office and the employees involved.

**Independent variables.** We measure technological proximity *(Tech)* between the collaborating organizations in year *t* based on the measurement proposed by Jaffe [62]. This measurement represents the extent to which two organizations' patents are included in the same technological classification. Following previous studies, we produce each organization's technology portfolio and then compute the uncentered correlations of their patent distribution vectors across technological classifications to measure the technological proximity. The multidimensional vector $P_{ik} = (P_i^1, P_i^2 \ldots P_i^k)$ can be used to capture the distribution where $P_i^k$ is the number of patents assigned to organization i in technological classification k. According to the vectorial angle approach, technological proximity between organization i and organization j is measured using the formula below:

$$Tech_{ij} = \frac{\sum_{k=1}^{K} P_{ik} P_{jk}}{\sqrt{\sum_{k=1}^{K} P_{ik}^2 \sum_{k=1}^{K} P_{jk}^2}}$$

where the vector $P_{ik}$ represents the number of patents of organization i in IPC class of *k*, and $P_{jk}$ represents the number of patent filed to organization *j* before the year of observation *t* in IPC class *k* respectively.

We measure geographical distance as a continuous positive variable using the spatial distance (expressed in Kilometres) between the location sites of the collaborating organizations (see S1 Appendix). The applicants' location is obtained using the name of the city of the applicants on the patent information. Then, the variable is standardized and subtracted from 0 to obtain geographical proximity *(Geo)*.

**Moderating variables.** Knowledge base decomposability *(Decomposability)* is measured using the global clustering coefficient of organizations' knowledge space in year *t*. First, an intra-organizational knowledge space is constructed for each of the sample organizations using the patents applied between *t-5~t*. Knowledge pace is essentially a network formed by different knowledge elements represented by IPCs. IPC is usually defined at one-, three- and four-digit levels. For example, C is the technological classification of "chemistry or metallurgy"; C23 represents the technological domain of "plating of metal materials; surface chemistry treatment; general coating methods; general inhibition of corrosion"; C23D represents the technological domain of "Enamel or glass coating of metal". We see that the four-digit level class of IPC can represent a narrow technological area which is suitable for our study.

Following Guan and Liu [46], we used the four-digit level class of IPC as a proxy for a knowledge element. If two knowledge elements indicated by technological classification codes jointly occur in a patent, we assume that there is a tie between them. Then, knowledge base decomposability is measured using the global clustering coefficient of the knowledge space. The measurement is calculated using the percentage of closed triads in the sum of closed and open triads in the network according to the formula below. Knowledge base decomposability of collaborating organizations is summed up and averaged to obtain the collaborative-dyad level measurement.

$$Clustering_{i,t-5-t} = \frac{3N_\Delta}{N_V}$$

Following the previous studies [19], the degree centrality of knowledge elements *(Degree centrality)* is measured by summing up the degree centrality of all the knowledge elements in the organizations' knowledge space. The degree centrality represents the number of direct social ties of a node in the network. We apply the normalized degree centrality as we compare values across graphs. The degree centrality of collaborating organizations is summed up and averaged to obtain the collaborative-dyad level measurement.

**Control variables.** We controlled for several other characteristics of the collaborating organizations that might influence the collaborative recombination innovation. All the variables are first measured on a single organization and then are summed and averaged over the partnering organizations to conduct analysis on a dyadic level. First, in the social network study, brokerage or structural hole position indicates the position of the actor between unconnected two actors, which gives the actor control advantage and access to heterogeneous knowledge. Similarly, we believe that positing between two unconnected knowledge elements, the focal knowledge element enjoys better recombination potential. Thus, in our analysis, the level of the structural hole of knowledge elements *(Structural hole)* in the knowledge space of collaborating organizations is controlled. It is obtained by the sum of the structural hole index of all the knowledge elements in organizations' knowledge base [19]. We obtain the structural hole measure based on Burt's [63] measure of constraints subtracted from 1. Lower values on this measure imply that knowledge elements occupy less constrained positions, thereby brokering more extensively in the network. We transformed the network constraint by subtracting it from 1 so that higher values indicate a larger structural hole. In addition, organizations with a high research intensity could have a stronger capability to recombine partner's knowledge. So, we control for past innovation performance of the collaborating organizations *(Past performance)* using the sum of patent count produced by the organizations in the previous five years. Then, the age of the collaborating organizations *(Age)* is controlled, which is obtained by the current year minus the earliest year the organization has filed a patent.

In addition, the level of utilization of the Internet might influence the effect of geographic proximity on collaborative innovation. In order to address this effect, we use the sum of Internet user percentage *(Internet)* of the collaborating organizations' countries as a control variable. Internet user percentages of different countries in 2001, 2006, 2010, 2014, and 2016 are obtained from the OECD Science, Technology, and Industry Scoreboard Report (OECD 2003, 2011, 2015, 2017), and are utilized to represent the internet usage of the sample organizations' country during 2000–2018. The team size of the collaborative innovation project *(Team size)* is obtained using the number of unique inventor names on the joint patent. Knowledge base variety *(Variety)* is measured using the number of IPC that appeared on the collaborating organizations' patents applied in the previous five years. Country *(Country)* is set to one if the collaborating organizations are from the same country and set to zero if the collaborating

organizations are from different countries. Different types of organizations may have different innovative patterns and influence the probability of recombination innovation. Organization type (*Organizational type*) controls for the organizational type of dyadic collaboration. It is a categorical variable representing whether the dyadic collaboration is between different firms or with universities and research institutions (1: firm-firm; 2: firm-university; 3: university-university; 4: university-research institution; 5: firm-research institution; 6: research institution-research institution). We also control for the number of knowledge elements in the knowledge space (*Knowledge element*). This measure is obtained using the count of unique knowledge components (IPCs) within the organization's knowledge space during the previous five years.

## Results

Table 1 presents the distribution of the number of patents and the number of technology classes in the year of observations by companies in the final sample. As shown, there are a few large corporations, universities, or research institutions with an accumulated patent number of around 300. Most of the organizations are small to medium organizations with an accumulated patent number of around 100. The technological classes vary across organizations ranging from 7 to 166.

Table 2 presents the distribution of the number of patents and the number of technology classes of collaborative dyad organizations. As shown, in the subgroup of co-patenting organizations that do not produce recombination innovation, the dyad organizations have the average patent number of 34 and 26 (Technological classes: 14 and 10). Comparably, in the subgroup of co-patenting organizations that produces recombination innovation, the dyad organizations have the average patent number of 13 and 13 (Technological classes: 7 and 20). It shows that, on average, specialized organizations tend to introduce new combinations in cooperation with other similar-sized organizations with diverse sets of technologies.

For the purpose of showing knowledge base characteristics more clearly, we construct the illustrative example knowledge spaces of the sample organizations. Fig 2A and 2B show different levels of knowledge base decomposability. Fig 2A is the knowledge space of Infineon Technologies North America Corporation with the knowledge base decomposability of 0.58, and

**Table 1. Distribution of patents and technological classes of sample organizations.**

| Accumulated patent number at year t | >300 | 200<x<300 | 150<x<200 | 100<x<150 | 50<x<100 | <50 |
|---|---|---|---|---|---|---|
| Number of organizations | 5 | 7 | 10 | 8 | 23 | 353 |
| Average technological classes | 166.80 | 83.71 | 34.76 | 26.62 | 22.78 | 7.72 |

**Table 2. Distribution of patents and technological classes of collaborative dyad organizations.**

| | recombination: 0 | | | | | recombination: 1 | | | | |
|---|---|---|---|---|---|---|---|---|---|---|
| | N | mean | S.D. | min | max | N | mean | S.D. | min | max |
| Accumulated patent number of organization 1 | 333 | 34.18 | 61.51 | 0 | 328 | 151 | 13.06 | 32.50 | 0 | 327 |
| Accumulated patent number of organization 2 | 333 | 26.26 | 53.96 | 0 | 482 | 151 | 13.16 | 25.11 | 1 | 204 |
| Technological classes of organization 1 | 333 | 14.26 | 22.68 | 1 | 204 | 151 | 7.44 | 10.84 | 0 | 111 |
| Technological classes of organization 2 | 333 | 10.59 | 9.34 | 0 | 43 | 151 | 20.6 | 46.71 | 0 | 326 |
| Dyadic Difference of accumulated patents | 333 | 39.34 | 66.74 | 0 | 470 | 151 | 8.72 | 23.33 | 0 | 200 |
| Dyadic Difference of technological classes | 333 | 10.74 | 20.30 | 0 | 177 | 151 | 22.64 | 52.91 | 0 | 328 |

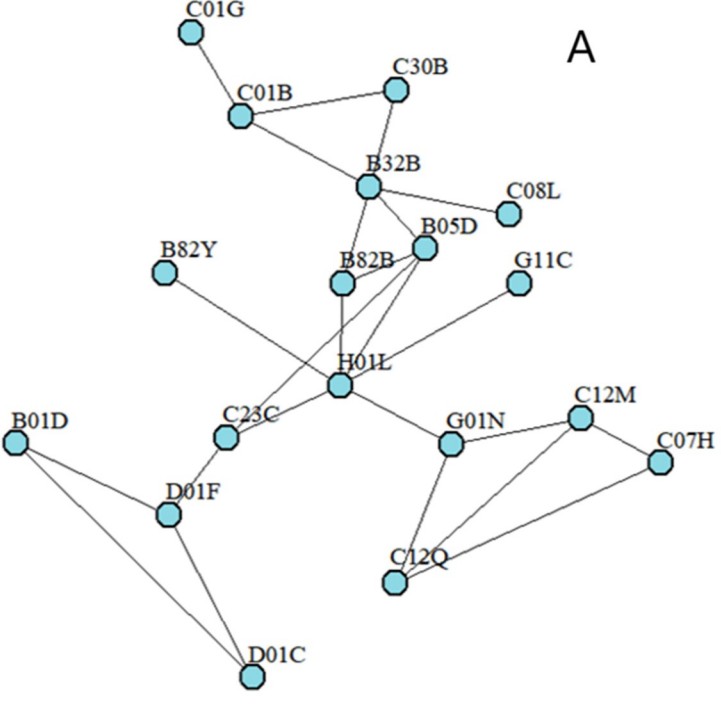

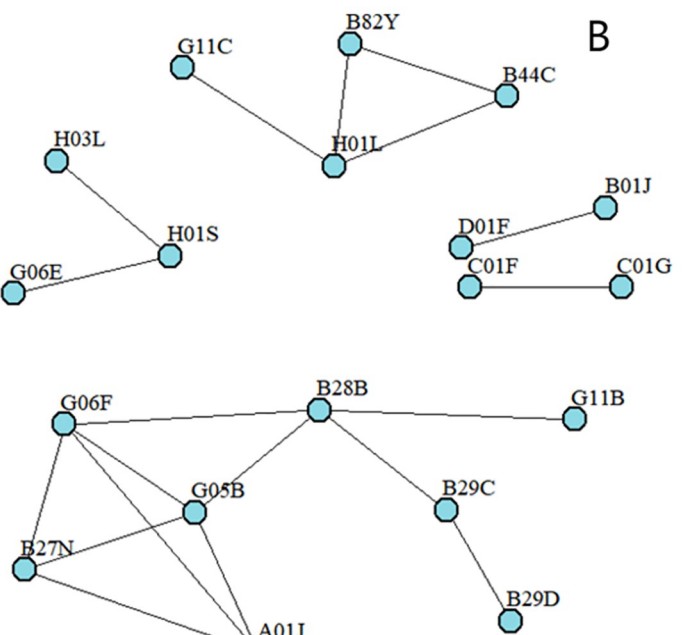

**Fig 2.** A. Illustration of the low knowledge network decomposability. B. Illustration of the high knowledge network decomposability.

Fig 2B is NEC Corporation with the knowledge base decomposability of 0.84. As shown, in the latter network, there are more closed triad relations compared to the former network. Fig 3A and 3B shows different levels of degree centrality of knowledge elements. Fig 3A is the knowledge space of International Business Machines Corporation with the knowledge element degree centrality of 19.83. Fig 3B is the knowledge space of Kabushiki Kaisha Toshiba with the knowledge element degree centrality of 47.5. As shown, in the latter network, knowledge components are connected to more knowledge components on average.

The dependent variable of the estimation is a binary variable. To determine whether we use fixed-effect or random-effect models, we conduct a Hausman test. The result of the Hausman test is 0.27 (above 0.05), so we employ random-effect panel Logit regression to conduct longitudinal analyses through STATA 12.0. Before employing the Logit model, we conduct several tests to ensure that the Logit model can be used to analyze the data. First, a VIF test is conducted before the estimation. The VIFs of variables are presented in Table 3 and the average of VIFs is below 3, eliminating the issue of multicollinearity. Second, we test the independence of observations among variables using the Durbin-Watson test. Durbin-Watson test can test whether there are first-order autocorrelations in the residuals and whether the model conforms to the assumption of independence. The results show that the observations of variables are independent of each other. Third, we test the linear relationship between explanatory variables and the logit of the response variable using OLS (Ordinary Least Squares) model. We use the scatter plot and linear fitting plot to verify the linear relation. The results show that there is a significant linear relation between explanatory variables and the logit of the response variable. In addition, we check whether there are outliers in the variables. Cook's distance in statistical analysis is often used to diagnose the presence of abnormal data in various regression analyses. The maximum Cook's Distance of the variables in our study is below 0.5 indicating that there are no outliers. For variables with high leverage cases, we winsorize the quantile of the continuous variables below 1% and above 99% of the explanatory variables before including them in the regression model.

We present the basic statistics in Table 3, and we report the correlations among the variables in Table 4. As shown in Table 3, the mean value of recombination innovation is 0.32, and the standard deviation is 0.47, which means that the capability of recombination innovation varies largely across different organizations. As shown in Table 4, the correlations between the variables are all below the threshold of 0.7, meaning that there is no multicollinearity issue.

In Table 5, Model 1 is the model incorporating all the control variables. As is seen, past performance is negatively related to the recombination innovation in collaboration. It means that organizations that have a larger number of accumulated patents do not perform well in the realm of recombination innovation. In Model 2, technological proximity is included, and the results show significant negative effect ($\beta$ = -1.606, $\rho$<0.001) indicating a linear negative relation between technological proximity and recombination innovation in collaboration. Hypothesis 1 is supported. In Model 3, geographic proximity and its quadratic term are included, and the results show significant negative effect of the quadratic term ($\beta$ = -0.441, $\rho$<0.05) indicating an inverted U-shaped relation between geographic proximity and recombination innovation in collaboration. Thus, Hypothesis 2 is supported.

In Model 4, the interaction term of knowledge base decomposability and technological proximity is included. As shown, the interaction term is negative and significant ($\beta$ = -9.175, $\rho$<0.05) supporting Hypothesis 3. In Model 5, the interaction term of decomposability and the quadratic term of geographic proximity is positive and significant ($\beta$ = 2.946, $\rho$<0.01), supporting Hypothesis 4. In Model 6, the interaction term of degree centrality of knowledge elements and technological proximity is included. The interaction term is positive and significant ($\beta$ = 0.162, $\rho$<0.01), supporting Hypothesis 5. In Model 7, the interaction of the degree

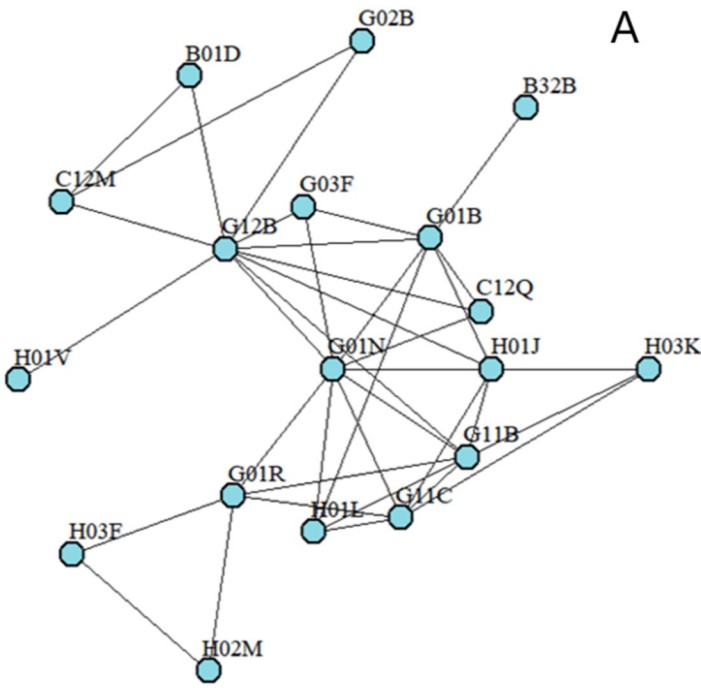

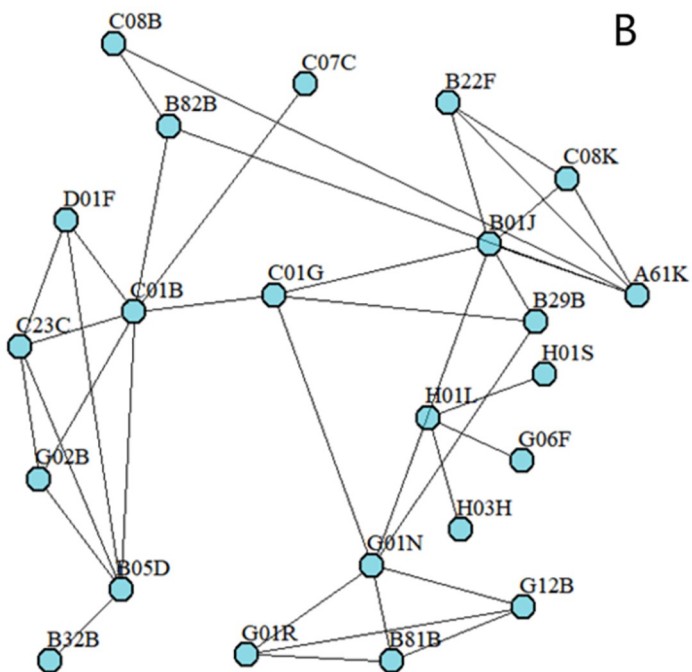

**Fig 3.** A. Illustration of the low degree centrality of knowledge elements. B. Illustration of the high degree centrality of knowledge elements.

**Table 3. Descriptive statistics of the variables.**

| Variable | Mean | Std. Dev. | Min | Max | VIF |
|---|---|---|---|---|---|
| Recombination | .32 | .47 | 0 | 1 | |
| Geo | 1 | 1 | -3.24 | 1.83 | 2.35 |
| Tech | .34 | .24 | 0 | 1 | 1.29 |
| Decomposability | 0.82 | .12 | .36 | 1 | 1.57 |
| Degree centrality | 33.03 | 8.48 | 3.61 | 48.6 | 3.31 |
| Structural hole | .87 | .09 | .35 | 0.94 | 2.85 |
| Past performance | 37.27 | 44.82 | 0 | 201.5 | 3.99 |
| Age | 10.93 | 5.00 | 1 | 26 | 1.90 |
| Internet | 72.47 | 15.42 | 20 | 91.5 | 1.48 |
| Team size | 4.26 | 2.32 | .08 | 16 | 1.14 |
| Variety | 16.12 | 15.10 | 2 | 111 | 3.81 |
| Country | 1 | 1.17 | 0 | 13 | 2.24 |
| Organizational type | 3.51 | 1.55 | 1 | 8 | 1.15 |
| Knowledge element | 17.82 | 11.26 | 4 | 61 | 1.16 |
| **Mean VIF** | | | | | **2.26** |

centrality of knowledge elements and the quadratic term of geographic proximity is positive and significant ($\beta = 0.097$, $\rho < 0.01$), supporting Hypothesis 6.

To provide a graphical representation of the moderating effect, we plot the moderation diagram in Figs 4A, 4B, 5A and 5B. The plots in Fig 4A and 4B more clearly show the moderating effect of knowledge base decomposability. As is shown in Fig 4A, technological proximity shows a negative linear effect, and the slope becomes steeper at a higher level of knowledge base decomposability indicating a negative moderation effect. In Fig 4B, geographic proximity shows a curvilinear effect, while the inverted-U shape flattens and becomes a positive linear

**Table 4. Correlations among the variables.**

| | 1 | 2 | 3 | 4 | 5 | 6 | 7 | 8 | 9 | 10 | 11 | 12 | 13 |
|---|---|---|---|---|---|---|---|---|---|---|---|---|---|
| 1. Recombination | 1 | | | | | | | | | | | | |
| 2. Geo | 0.04 | 1 | | | | | | | | | | | |
| 3. Tech | 0.16 | 0.04 | 1 | | | | | | | | | | |
| 4. Decomposability | 0.31* | 0.05 | 0.09 | 1 | | | | | | | | | |
| 5. Degree | -0.03 | -0.05 | 0.25* | -0.16 | 1 | | | | | | | | |
| 6. Structural hole | 0.02 | -0.116 | 0.18* | -0.17 | 0.67* | 1 | | | | | | | |
| 7. Past performance | -0.22* | -0.10 | -0.33* | -0.46* | -0.15 | 0.03 | 1 | | | | | | |
| 8. Age | -0.17 | -0.28* | 0.13 | -0.36* | 0.15 | 0.21* | 0.26* | 1 | | | | | |
| 9. Internet | -0.12 | -0.02 | 0.16 | -0.19* | 0.35* | 0.23* | 0.08 | 0.44* | 1 | | | | |
| 10. Team size | -0.06 | -0.03 | 0.09 | -0.12 | 0.08 | 0.03 | -0.02 | 0.15 | 0.19* | 1 | | | |
| 11. Variety | -0.15 | -0.13 | -0.25* | -0.44* | 0.02 | 0.10 | 0.61* | 0.41* | 0.19* | -0.06 | 1 | | |
| 12. Country | -0.05 | 0.70* | 0.09 | -0.07 | 0.02 | -0.06 | -0.07 | -0.09 | -0.02 | -0.11 | -0.05 | 1 | |
| 13. Organizational type | 0.10 | 0.10 | 0.10 | -0.00 | 0.00 | 0.05 | -0.15 | -0.19* | -0.21* | -0.09 | -0.17 | 0.12 | 1 |
| 14. Knowledge element | -0.19* | -0.04 | -0.29* | -0.46* | -0.04 | 0.11 | 0.60* | 0.24* | 0.08 | 0.02 | 0.59* | 0.01 | -0.09 |

*$p < 0.05$,

**$p < 0.01$,

***$p < 0.001$ (same below)

**Table 5. Regression results.**

| | Model 1 | Model 2 | Model 3 | Model 4 | Model 5 | Model 6 | Model 7 |
|---|---|---|---|---|---|---|---|
| Tech | | -1.606** | -2.684*** | -1.896* | -2.808*** | -3.837*** | -3.026*** |
| | | (0.761) | (0.905) | (1.077) | (1.011) | (1.137) | (1.136) |
| Geo | | | 1.320** | 0.939** | 2.477*** | 0.815** | 3.081*** |
| | | | (0.640) | (0.368) | (0.720) | (0.350) | (0.784) |
| Geo^2 | | | -0.441* | | -1.270*** | | -1.819** |
| | | | (0.286) | | (0.350) | | (0.458) |
| Decomposability | | | | 2.629** | 3.334** | | |
| | | | | (1.327) | (1.463) | | |
| Tech* Decomposability | | | | -9.175* | | | |
| | | | | (4.960) | | | |
| Geo* Decomposability | | | | | -2.860 | | |
| | | | | | (1.521) | | |
| Geo^2* Decomposability | | | | | 2.946** | | |
| | | | | | (1.098) | | |
| Degree centrality | | | | | | -0.061* | -0.240*** |
| | | | | | | (0.033) | (0.063) |
| Tech* Degree centrality | | | | | | 0.162** | |
| | | | | | | (0.068) | |
| Geo* Degree centrality | | | | | | | -0.036* |
| | | | | | | | (0.025) |
| Geo^2* Degree centrality | | | | | | | 0.097** |
| | | | | | | | (0.020) |
| Structural hole | 0.648 | 0.344 | 0.451 | 2.290 | 2.851 | 5.731 | -3.875 |
| | (0.913) | (0.907) | (3.579) | (3.773) | (4.127) | (4.166) | (4.842) |
| Past performance | -0.010*** | -0.010** | -0.022*** | -0.019*** | -0.013** | -0.024*** | -0.023*** |
| | (0.004) | (0.004) | (0.006) | (0.006) | (0.006) | (0.006) | (0.007) |
| Age | -0.032 | -0.041* | -0.065** | -0.037 | -0.037 | -0.098*** | -0.067* |
| | (0.024) | (0.024) | (0.028) | (0.032) | (0.031) | (0.031) | (0.038) |
| Internet | -0.006 | -0.006 | -0.018** | -0.015 | -0.016* | -0.013 | -0.021* |
| | (0.006) | (0.006) | (0.009) | (0.009) | (0.009) | (0.009) | (0.011) |
| Team size | -0.032 | -0.041 | -0.221** | -0.211* | -0.091 | -0.211** | -0.294** |
| | (0.077) | (0.078) | (0.104) | (0.120) | (0.142) | (0.100) | (0.134) |
| Variety | 0.018 | 0.021* | 0.050*** | 0.049*** | 0.037** | 0.059*** | 0.051*** |
| | (0.012) | (0.012) | (0.016) | (0.015) | (0.016) | (0.017) | (0.019) |
| Country | -0.723 | -0.863 | -4.489*** | -3.212*** | -4.513*** | -4.000*** | -6.192*** |
| | (0.544) | (0.536) | (1.353) | (1.212) | (1.408) | (1.257) | (1.898) |
| Organizational type | 0.075 | 0.055 | 0.060 | 0.210 | 0.178 | 0.129 | 0.043 |
| | (0.113) | (0.113) | (0.151) | (0.157) | (0.166) | (0.150) | (0.167) |
| Knowledge element | 0.020 | 0.061 | 0.060 | 0.120 | 0.130 | 0.090 | 0.060 |
| | (0.038) | (0.037) | (0.043) | (0.190) | (0.056) | (0.049) | (0.043) |
| Year | Included | Included | Included | Included | Included | Included | Included |
| Constant | 0.212 | 0.567 | 5.378 | -3.778 | 0.989 | 1.195 | 14.563** |
| | (1.777) | (1.752) | (3.489) | (4.624) | (4.005) | (3.765) | (7.415) |
| Observations | 350 | 350 | 350 | 350 | 350 | 350 | 350 |
| N | 152 | 152 | 152 | 152 | 152 | 152 | 152 |
| Wald chi2 | 16.32 | 19.94 | 42.17 | 46.38 | 48.47 | 41.62 | 27.18 |
| Prob>chi2 | 0.038 | 0.018 | 0.042 | 0.021 | 0.017 | 0.061 | 0.061 |

(*Continued*)

**Table 5.** (Continued)

| | Model 1 | Model 2 | Model 3 | Model 4 | Model 5 | Model 6 | Model 7 |
|---|---|---|---|---|---|---|---|
| Log likelihood | -129.03 | -126.668 | -93.88 | -89.98 | -84.74 | -91.36 | -90.450 |

*p<0.05,

**p<0.01,

***p<0.001 (same below)

effect at a higher level of knowledge base decomposability indicating a positive moderation effect [56].

As shown in Fig 5A, the slope of the negative linear effect of technological proximity becomes smaller at a higher level of degree centrality of knowledge elements indicating a positive moderating effect. In Fig 5B, the inverted-U shape flattens, and the optimal point moves to the right at a higher degree centrality indicating a positive moderation effect.

## Robustness check

We utilized an alternative measure of knowledge base decomposability in the robustness check. When measuring the clustering coefficient of the knowledge space, instead of using the global clustering coefficient, we utilized the ratio of the global clustering coefficient of the knowledge space to the clustering of a random network with the same number of vertices and edges. In addition, in the measurement of control variables, instead of using the average values of the dyad organizations, we utilized the summarized values and alongside controlled for the absolute differences. The regression results are displayed in Table 6. As shown, the results show similar effects of the independent variables except that the moderation effect of decomposability on the relation between technological proximity and recombination innovation is not significant. The sign of the coefficient is same as the main results. The possible explanation is that the measurement of the global network clustering coefficient comparing to the random network usually has a smaller variance as it takes into account network size and density. As a result, the moderation effect of knowledge base decomposability is less obvious using this alternative measure.

We also utilized an alternative measure of recombination innovation. When collaborative organizations create a patent that contains an IPC code that is new to both of them, they are collaborating to learn something that is completely new to them. We remove cases of such recombination innovation to obtain an alternative measure. For example, organization A possesses IPC class X and Z and organization B possesses IPC class Y and K. If organization A and B collaborate and create patent 1 with IPC class X, Y and patent 2 with IPC class M, then patent 1 is considered recombination innovation while patent 2 is not in this measurement. In the regression results with the alternative measure of recombination innovation, all of the main effects stay the same.

## Discussion and conclusions

Recombination innovation is the process of combining existing knowledge elements in a new configuration to create innovative value. Previous research emphasized the origin and value of recombination innovation while mostly studying it at individual organizational level. However, few studies address the question of what factors influence recombination innovation at collaborative dyad level. In contrast to the previous studies, this study focused on the recombination innovation collaboratively developed by collaborative agents and investigated the joint

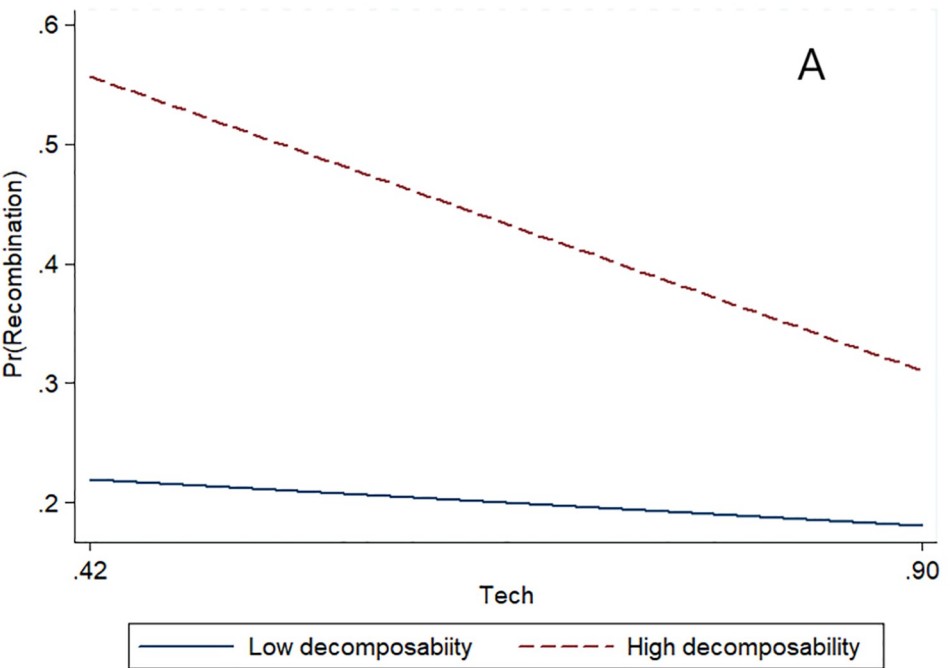

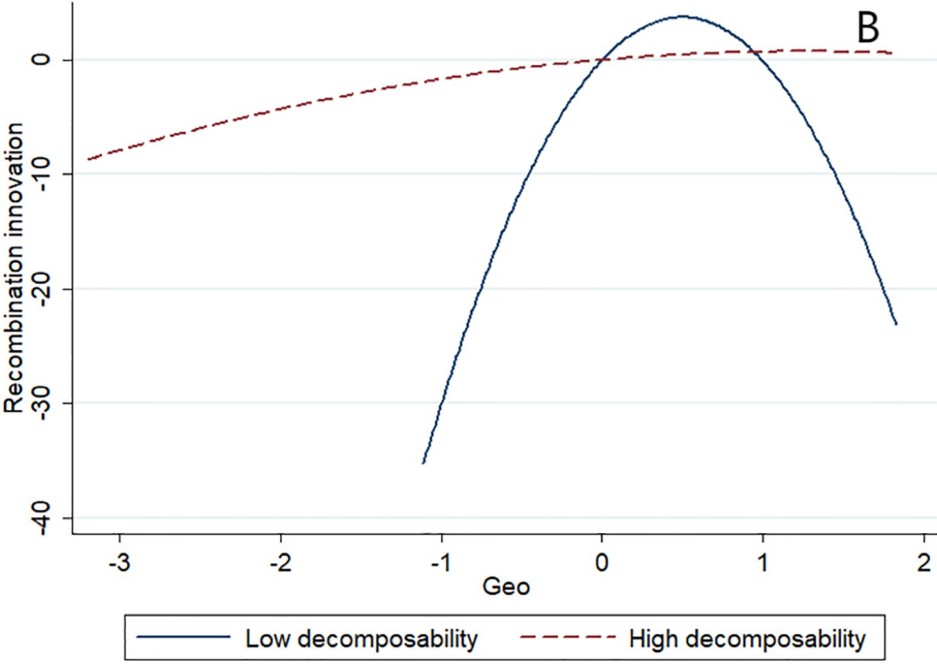

**Fig 4.** A. Moderating effect of knowledge base decomposability on the relationship between technological proximity and recombination innovation. B. Moderating effect of knowledge base decomposability on the relationship between geographic proximity and recombination innovation.

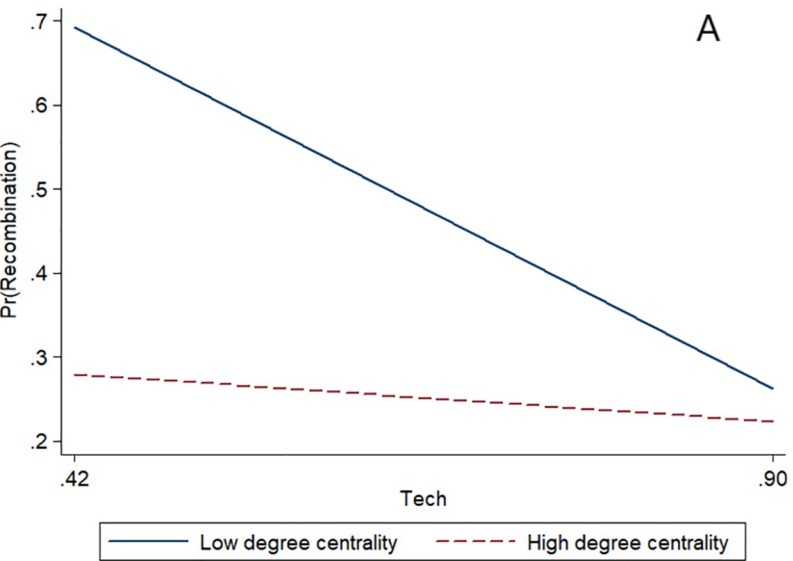

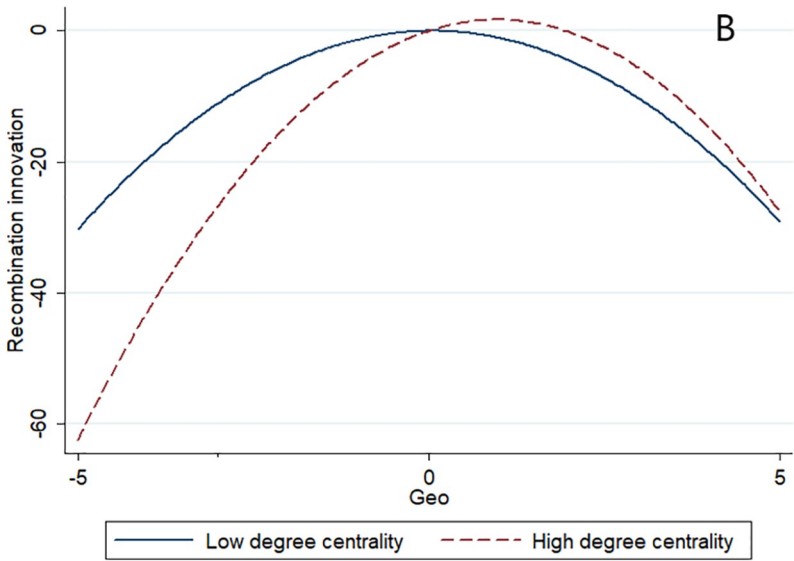

**Fig 5.** A. Moderating effect of degree centrality of knowledge elements on the relationship between technological proximity and recombination innovation. B. Moderating effect of degree centrality of knowledge elements on the relationship between geographic proximity and recombination innovation.

effect of proximity and knowledge base characteristic of partners on this process. Based on the longitudinal analysis of the nanotechnology industry from 2000–2018, we come to several conclusions.

We have found that there is a negative relation between technological proximity and recombination innovation in R&D collaboration, which is different from the inverted U-shaped relation put forward in the previous studies. The result shows that in seeking recombination innovation during external collaboration, novelty plays a more influential effect than

**Table 6. Regression results using alternative measures of decomposability and control variables.**

| | Model 1 | Model 2 | Model 3 | Model 4 | Model 5 | Model 6 | Model 7 |
|---|---|---|---|---|---|---|---|
| Tech | | -2.388** | -2.024* | -2.254* | -4.081** | -4.219*** | -1.799 |
| | | (1.129) | (1.131) | (1.351) | (1.620) | (1.426) | (1.594) |
| Geo | | 0.698* | 1.049* | 0.889** | 1.977 | 0.687* | 3.294*** |
| | | (0.366) | (0.600) | (0.432) | (1.237) | (0.381) | (0.837) |
| Geo^2 | | | -0.498* | | -0.907 | | -2.355*** |
| | | | (0.279) | | (0.578) | | (0.521) |
| Decomposability | | | | 1.256 | -7.273** | | |
| | | | | (1.280) | (3.565) | | |
| Tech* Decomposability | | | | -7.980 | | | |
| | | | | (5.134) | | | |
| Geo* Decomposability | | | | | -0.000 | | |
| | | | | | (0.002) | | |
| Geo^2* Decomposability | | | | | 3.358* | | |
| | | | | | (1.845) | | |
| Degree centrality | | | | | | -0.065* | -0.313*** |
| | | | | | | (0.034) | (0.077) |
| Tech* Degree centrality | | | | | | 0.220*** | |
| | | | | | | (0.084) | |
| Geo* Degree centrality | | | | | | | -0.023 |
| | | | | | | | (0.024) |
| Geo^2* Degree centrality | | | | | | | 0.118*** |
| | | | | | | | (0.028) |
| Structural hole | 1.085 | 0.715 | 0.865 | 2.200 | 4.521 | 4.707 | 4.550 |
| | (0.941) | (3.760) | (3.810) | (3.924) | (4.273) | (4.377) | (4.792) |
| Past performance_Sum | -0.002 | -0.021*** | -0.023*** | -0.024*** | -0.017** | -0.023*** | -0.053*** |
| | (0.005) | (0.008) | (0.008) | (0.009) | (0.008) | (0.009) | (0.013) |
| Age_Sum | -0.030 | -0.068** | -0.083** | -0.030 | -0.056 | -0.107*** | -0.146*** |
| | (0.024) | (0.033) | (0.035) | (0.038) | (0.040) | (0.038) | (0.054) |
| Internet_Sum | -0.012* | -0.023* | -0.039** | -0.015 | -0.032* | -0.022 | -0.059*** |
| | (0.007) | (0.013) | (0.016) | (0.014) | (0.017) | (0.015) | (0.021) |
| Variety_Sum | 0.019 | 0.074*** | 0.072** | 0.089*** | 0.085** | 0.078*** | 0.105*** |
| | (0.016) | (0.026) | (0.028) | (0.030) | (0.036) | (0.029) | (0.038) |
| Team size | -0.018 | -0.187* | -0.197* | -0.204 | -0.093 | -0.202* | -0.467*** |
| | (0.076) | (0.109) | (0.111) | (0.133) | (0.156) | (0.110) | (0.178) |
| Past performance_Diff | -0.025*** | -0.007 | -0.003 | -0.003 | 0.002 | -0.009 | 0.011 |
| | (0.009) | (0.011) | (0.011) | (0.013) | (0.013) | (0.011) | (0.016) |
| Age_Diff | 0.114** | 0.122** | 0.113* | 0.099 | 0.125* | 0.113* | 0.160** |
| | (0.047) | (0.060) | (0.061) | (0.063) | (0.067) | (0.063) | (0.073) |
| Internet_Diff | -0.090** | -0.023 | -0.129 | 0.044 | -0.071 | -0.053 | -0.185* |
| | (0.040) | (0.070) | (0.090) | (0.073) | (0.095) | (0.076) | (0.105) |
| Variety_Diff | -0.000 | -0.048 | -0.044 | -0.058 | -0.061 | -0.081* | -0.085 |
| | (0.025) | (0.042) | (0.040) | (0.041) | (0.040) | (0.048) | (0.055) |
| Country | -1.279* | -2.740** | -4.688*** | -2.039 | -4.466** | -3.705*** | -9.889*** |
| | (0.662) | (1.248) | (1.728) | (1.321) | (2.004) | (1.417) | (2.358) |
| Organizational type | 0.162 | 0.225 | 0.121 | 0.339* | 0.307 | 0.199 | -0.093 |
| | (0.121) | (0.170) | (0.174) | (0.187) | (0.201) | (0.174) | (0.207) |

(*Continued*)

**Table 6.** (Continued)

| | Model 1 | Model 2 | Model 3 | Model 4 | Model 5 | Model 6 | Model 7 |
|---|---|---|---|---|---|---|---|
| Knowledge element | 0.021 | 0.060 | 0.040 | 0.100 | 0.120 | 0.080 | 0.070 |
| | (0.037) | (0.035) | (0.033) | (0.170) | (0.066) | (0.079) | (0.063) |
| Year | Included | Included | Included | Included | Included | Included | Included |
| Constant | 0.197 | 2.510 | 7.783* | -1.590 | 3.936 | 2.521 | 37.110*** |
| | (1.859) | (3.892) | (4.658) | (4.216) | (6.068) | (4.604) | (9.324) |
| Observations | 350 | 350 | 350 | 350 | 350 | 350 | 350 |
| N | 152 | 152 | 152 | 152 | 152 | 152 | 152 |
| Wald chi2 | 28.32 | 48.13 | 42.99 | 39.16 | 41.93 | 38.99 | 38.68 |
| Prob>chi2 | 0.005 | 0.054 | 0.0784 | 0.260 | 0.266 | 0.2184 | 0.309 |
| Log likelihood | -117.03 | -79.469 | -91.23 | -84.84 | -77.51 | -85.23 | -73.138 |

absorptive capacity. In line with the previous studies, we have also found that there is an inverted U-shaped relation between geographic proximity and recombination innovation in R&D collaboration.

In addition, going beyond the baseline hypotheses, we have found that knowledge base decomposability negatively moderates the effect of technological proximity as it enhances organizations' capability of exploring new knowledge. Conversely, knowledge base decomposability positively moderates the effect of geographic proximity as the cost of geographic distance is mitigated when knowledge base decomposability is high, and the knowledge homogeneity effect caused by large geographic proximity is also mitigated. The level of degree centrality of knowledge elements positively moderates the relation between technological proximity and recombination innovation, as it promotes organizations' usage of existing technologies while inhibiting their exploration of new technologies. The degree centrality of knowledge elements positively moderates the relation between geographic proximity and recombination innovation. More specifically, the benefits of geographic distance are enhanced, and the drawback of geographic proximity is mitigated. Also, the optimal geographic proximity would be larger when the degree centrality of knowledge elements is high.

## Theoretical implications

This paper contributes to the recombinant search and the economic geography literature. Economic geography literature investigates the effectiveness and output of collaborative innovation from the perspective of proximity while viewing organizations as a "black box", neglecting the nature of the recombination search process. The recombinant search studies have a long tradition of studying knowledge recombination process [64], while on the other hand, this literature stream tends to confine its theorizing to within-organization contexts, largely ignoring knowledge recombination across organizational boundaries. In this paper, focusing on the organization-to-organization collaborative dyad level, we bring relational characteristics, i.e., proximity and organizational characteristic, i.e., knowledge base characteristics into one analytical framework.

Our results indicate that proximity affects the emergence of novel combinations of knowledge during R&D collaborations. Previous studies show that collaborating partners need technological proximity to absorb partners' knowledge effectively, but too much proximity would lead to the knowledge homogeneity issue [30, 33]. Our study adds to the understanding of how organizations should balance absorptive capacity and technological variety when

collaborating with external partners. Our study indicates that technological variety outweighs the absorptive capacity during the recombination innovation process.

Previous studies show that geographical co-location of actors facilitates localized learning and lays the foundation for technological recombination [65]. Our study adds to this line of research by stating that a moderate level of geographical proximity increases the likelihood that actors learn from each other and also have access to distinct but related technological knowledge. In addition, our study conforms to Juhász et al.'s [65] study stating that a moderate level of geographic proximity between organizations with specific knowledge of technologies increases the chances that connections may be discovered between previously unrelated technologies and that complementarities are identified.

Our results also indicate that important differences exist concerning the knowledge space and would influence how much learning effect and recombinant potential collaborative partners have at different levels of proximity. We manage to identify and theorize on novel mechanisms that can hamper or promote organizations' ability to benefit from external collaborations. We demonstrate that unique structural characteristics of knowledge linkages within organizations' knowledge pool can create substantial differences in their capability, technologies, and routines that would interact with different dimensions of proximity with partners and bring learning opportunities and challenges during collaborations.

In addition, our study adds to the existing literature that links research on inter-organizational collaboration and knowledge spaces. Specifically, by examining the role of knowledge space, we untangle the underlying mechanism of how organizations use their existing knowledge base to undertake a recombinant search in R&D collaboration. Previous studies emphasize the importance of an organization's knowledge base in determining the organizations' boundaries [66] and its capability to draw benefits from external collaborations. These studies mostly view the knowledge base as a collection of innovation resources or patents, while its structural characteristic is rarely analyzed. Shifting the conceptual lens from technological components to the interdependence or structure of knowledge elements, we apply the idea of "knowledge space" at the organizational level and link it to the outcomes of cooperation. We demonstrate that there are important implications at interdependence or structure of knowledge components in organizations' knowledge base, and the heterogeneity in knowledge linkages in knowledge space has implications that go beyond the boundary of organizations.

Specifically, during R&D collaborations, when partners' knowledge base has a high level of decomposability, it would influence their capability to collaborate across a large technological distance and geographic distance. This means that it would enhance their ability to absorb new knowledge from partners. When the degree centrality of the knowledge base in the knowledge space is high, it promotes organizations' capacity to conduct recombination innovation with partners that have similar technologies while also promoting their ability to learn from geographically distant partners. These findings suggest that organizations' internal knowledge space is an essential factor that explains the difference in their motivation and ability to find recombination opportunities during R&D collaborations. Previous studies have examined knowledge bases structural index such as homogeneity [23], recombination novelty [24], complementary and substitutability [67], structural hole [46], and interdependence [17]. In this paper, we point out the characteristic of decomposability, degree centrality and their interactive effect with proximity on joint recombination innovation.

## Managerial implications

Our study also has managerial implications for organizations choosing R&D collaborative partners. Apart from deciding on the proximity dimensions of inter-organizational

collaboration, organizations should pay attention to their and potential partners' internal knowledge base. To evaluate the knowledge base, organizations could inspect how cohesive the knowledge base is and to what extent their technologies are linked with other technologies. By jointly considering the proximity and knowledge base of the partners, organizations can make better decisions and achieve more effective learning through R&D collaborations.

## Limitations

There are limitations in our study. First, the sample of the study is organizations that successfully apply for joint patents. This would naturally exclude organizations that participate in other types of R&D collaboration. However, previous studies stated that joint patenting can be seen as the result of enduring R&D collaboration between organizations [65, 66]. We believe that our results show reference for most of the research-intensive organizations that tend to patent their research achievements. Second, our study does not differentiate between different forms of organizations like firms, universities, and research institutions. Future studies can address different organizational forms and examine whether this would affect the recombination output during collaborations.

## Supporting information

**S1 Appendix. Calculation of the geographic distance between two cities.**
(DOCX)

## Author Contributions

**Conceptualization:** Ding Nan.

**Data curation:** Ding Nan.

**Formal analysis:** Ding Nan.

**Methodology:** Ding Nan.

**Writing – original draft:** Ding Nan.

**Writing – review & editing:** Ding Nan.

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
