## [Decision Letter · Decision Letter 0]

15 Mar 2023

PONE-D-22-33904The influence of proximity and knowledge base on recombination innovation in R&D CollaborationPLOS ONE

Dear Dr. Nan,

Thank you for submitting your manuscript to PLOS ONE. After careful consideration, we feel that it has merit but does not fully meet PLOS ONE’s publication criteria as it currently stands. Therefore, we invite you to submit a revised version of the manuscript that addresses the points raised during the review process.

We look forward to receiving your revised manuscript.

Kind regards,

Claudia Noemi González Brambila, Ph.D.

Academic Editor

PLOS ONE

Journal Requirements:

3. We note that you have stated that you will provide repository information for your data at acceptance. Should your manuscript be accepted for publication, we will hold it until you provide the relevant accession numbers or DOIs necessary to access your data. If you wish to make changes to your Data Availability statement, please describe these changes in your cover letter and we will update your Data Availability statement to reflect the information you provide

4. PLOS requires an ORCID iD for the corresponding author in Editorial Manager on papers submitted after December 6th, 2016. Please ensure that you have an ORCID iD and that it is validated in Editorial Manager. To do this, go to ‘Update my Information’ (in the upper left-hand corner of the main menu), and click on the Fetch/Validate link next to the ORCID field. This will take you to the ORCID site and allow you to create a new iD or authenticate a pre-existing iD in Editorial Manager. Please see the following video for instructions on linking an ORCID iD to your Editorial Manager account: https://www.youtube.com/watch?v=_xcclfuvtxQ.

Reviewers' comments:

Reviewer's Responses to Questions

**Comments to the Author**

1. Is the manuscript technically sound, and do the data support the conclusions?

Reviewer #1: Yes

Reviewer #2: Yes

2. Has the statistical analysis been performed appropriately and rigorously? 

Reviewer #1: Yes

Reviewer #2: Yes

3. Have the authors made all data underlying the findings in their manuscript fully available?

Reviewer #1: No

Reviewer #2: No

4. Is the manuscript presented in an intelligible fashion and written in standard English?

Reviewer #1: Yes

Reviewer #2: Yes

5. Review Comments to the Author

Reviewer #1: Summary

In their paper, the authors investigate the effects of proximity (geographical and technical) on the recombination innovation at the dyad level between companies. Additionally, the authors test how this effect is mediated by an organization's knowledge base structure. Thus, based on theories from the R&D field, the authors propose a theoretical framework that explains the interactive effect of proximity and knowledge network structure on companies' collaborative recombination innovation potential. They derive six hypotheses that test for the direct and interactional effects of the aforementioned variables. For their empirical evaluation, the authors utilize longitudinal patent data from 150 organizations in the global nanotechnology industry. After conducting their statistical analyses, the authors conclude that technological proximity exerts a negative effect on the recombination innovation of companies while geographic proximity has an inverted U-shaped effect. Moreover, they were able to show that an organization's knowledge base decomposability plays a negative role in moderating the effect of technological proximity while playing a positive role in regulating the effect of geographic proximity. Degree centrality on the other side positively moderates the effect of both technological and geographic proximity.

In this review, I will highlight the research’s strengths and weaknesses and outline the challenges and opportunities presented in the manuscript.

I enjoyed reading the manuscript and appreciate the authors’ work. They succeed in tackling an interesting subject by developing a comprehensive theoretical framework and conducting thorough empirical analyses. By discussing different potential effects of the indicators under scrutiny, the authors derive six plausible hypotheses and test them convincingly in their empirical analyses. Additionally, the included figures and tables are clearly arranged and help the reader to follow their conceptual as well as empirical procedure. Overall, the authors use reasonable language, which makes it possible to follow their presented ideas most of the time.

While the paper succeeds in delivering its main messages, my review reveals potential issues that might diminish the contribution of the presented manuscript:

1. No direct research question

In their introduction, the authors mention that they aim to analyze "how knowledge interdependence at the micro level and partners’ knowledge origin interact leading to the change in state of collaborative recombination innovation." Although they mention their research goal, it would be helpful if they could include precise research questions which would make it easier for the reader to understand the overall goal of the paper.

2. Introduction of theories

When introducing their concept of proximity, the authors mention the analytical framework of Boschma with its five dimensions of proximity. Yet, they never explain those five dimensions in detail. To help the reader understand their theoretical framework more thoroughly, the authors should introduce their utilized theories in more detail.

3. Operationalization of variables

It has to be positively emphasized that the authors operationalize all mentioned variables by including a measurement description thereof. Yet, in several cases (e.g. technological proximity or control variables like structural hole) the authors do not explain how the measurement relates to the theoretical concept. It would be helpful if the authors could highlight the link between concept and measurement more thoroughly. When mentioning their formula to calculate the spatial distance, the authors do not explain the meaning of the number "6357." Please explain this formula in more detail. Additionally, if the authors establish a link between concept and measurement (e.g. knowledge base decomposability), the paper could be enhanced if the authors could provide more information regarding the theoretical background (e.g. what is the four-digital level class of IPC?).

4. Creation of dataset

When curating their dataset, the authors mention that they exclude patents that are filed to individual inventors. Yet, they do not support this decision with any arguments. Likewise, the authors state that they aim to incorporate an organization's history by only including organizations that had at least one patent application before the observation year. In this case, the authors provide a justification for their decision, yet it would be helpful if they could elaborate on why it is important to take an organization's history into account and why they decided for this specific period.

5. Statistical analysis

Although the authors mention that they test for potential multicollinearity with the variance inflation factor (VIF), they do not control for any other assumptions necessary to conduct a logistic regression (e.g. independence of observations, no outliers, linear relationship between explanatory variables and the logit of the response variable). To obtain sound results, the authors should check for these assumptions since they could affect their empirical results and thus their theoretical contribution. Additionally, it would be helpful if the authors could include the R-squared value and the F1 score. Otherwise, it is hard to determine how suitable the model is for their data (how much variance in the dependent variable can be explained by the model?). Currently, the authors perform aggregated analyses with their data ranging from 2000-2018. To receive a more dynamic understanding of the relationship between the different variables, it would be interesting to perform longitudinal analyses since the authors work with time-series data.

6. Exemplary issues regarding language and format:

Grammar and wording:

- In a lot of cases, the authors forgot to mention the respective article in front of a noun (e.g. "Then, in the purpose of opening the “black box” of <the>relation between proximity and recombination innovation, we viewed knowledge base as a structural factor during <the> technological adaptation process among organizations."); please check the entire manuscript.

- p.1: "While to date, our comprehension of how partnering organizations access a broader knowledge pool, uncover novel perspectives, and spot recombination opportunities on the technological landscape is limited [6, 7]."; the word "while" needs to be deleted, otherwise one would expect a counter-argument at the end of the sentence.

- p.2: "Notably, existing studies has highlighted the importance of organizations’ external R&D collaboration as a critical mechanism to recombine novel knowledge from partners, the process of which would spur more recombination innovation."; the "has" needs to be changed into "have."

- p.2: I do not understand the following sentence: "Knowledge network represents the coupling logic of organizations and guilds organizations knowledge searching process."; please rewrite this sentence.

- p.2: "Previous studies puts forward that knowledge base is a network of knowledge element in essence [6, 19, 21, 22] that has great implications for formation of collaborations [20, 23], collaborative effectiveness [18, 20], and learning of partners’ knowledge [9]."; "puts" needs to be transformed into "put", "element" needs to be the plural form, "has" needs to be changed to "have", and there is an article missing before formation.

- p.3: "We detailly concentrate on two marked aspects of knowledge network which are knowledge base decomposability i.e., the extent to which knowledge elements are connected with each other forming clustered structure, and degree centrality of knowledge elements i.e., the extent to which knowledge elements are connected to other knowledge elements within and beyond organizational boundary."; to my knowledge, the word "detailly" does not exist, please revise. Additionally, "structure" should be transformed into the plural form.

- p.7: "While, too much similarities in knowledge base would hurt the alternative knowledge that can be considered limiting the recombination scope [47]."; the authors should delete the "while" and "much" needs to be changed to "many."

- p.9: "Therefore, locating in a vicinity of collaborating partners always means a high level of relationship quality, promotes the mutual understanding, and supports creativity and information exchange."; "locating in a vicinity" is hard to understand, simply rewrite it to e.g., "being spatially close to collaborating partners."

- p.20: "It means that large organizations and large research teams do not well perform in the realm of recombination innovation."; please change "well perform" to "perform well."

References

- p.26: "However, previous studies stated that joint patenting can be seen as the result of enduring R&D collaboration between organizations."; please provide some sources when mentioning "previous studies."

Format

- It seems like formulas were included with a screenshot and are therefore of low-resolution quality. Please check and insert them directly in the document. If possible, also increase the resolution of the inserted figures.

7. Conclusion

In summary, I can state that the authors pursue an interesting line of research that highlights the importance and different effects of proximity on the recombination innovation of companies on a dyad level. Yet, there are some issues, particularly regarding their operationalization of variables and their statistical analyses that could be improved. I hope my suggestions are helpful and wish them success as they continue their research.</the></the>

Reviewer #2: The paper illustrates the connection between proximity of firms and the creation of recombination-based patents at the dyad-level. Proximity of firms is considered through geographic proximity, technological proximity and the similarity of company-level knowledge bases.

I find the basic idea behind the paper interesting, however, I have a list of comments and suggestions for improvement below:

1. The first sentence of the abstract is very hard to follow. I would suggest to please the readers and catch their attention with an easy-to-understand beginning.

2. The applied methodology is not mentioned/hinted in the abstract at all.

3. The Proximity, knowledge base and recombination innovation section of the paper does not refer to the growing literature about atypical combinations and teamwork. I would suggest to build on this stream of literature to strengthen the motivation of the paper. Why are recombination innovation important in general?

Uzzi et al. (2013) Atypical combinations and scientific impact. Science 342 (6157), 468-472.

4. In general, the paper would benefit from descriptive figures that illustrate the sample and the key concepts. For example, network metrics are applied, so an illustrative example network (or two, since the focus is on dyads) could make the paper more appealing and easier to follow. It would be nice to see the distribution of the number of patents and the number of technology classes by companies in the final sample.

6. Additionally, illustration on how balanced are dyads in terms of patents by company and IPC classes filed by company would be nice to see. Is it the case that large, productive and often specialized companies tend to introduce new combinations in cooperation with smaller, more innovative, but less experienced firms? In such a situation, a small firm with a single patent would have a highly clustered knowledge network by definition. Clear descriptive on this regard would help.

7. The dependent variable (recombination innovation) only considers combinations that are new to the two companies in the focal dyad. Since patents represent highly codified innovation, firms can put resources together to more effectively copy an already existing solution (or moved towards a technological solution). Could you illustrate how often the observed dyads introduce new combinations that are novel for the entire sample of companies? Focusing only on absolute novel combinations might lead to a too small sample size in your context (?), however, discussing your results in relation to other studies highlighting the role of geography for the first combination of technology classes on patents might improve the paper.

Juhász et al. (2020) Explaining the dynamics of relatedness: The role of co-location and complexity. Papers in Regional Science 100 (1), 3-21.

8. The dependent variable (recombination innovation) needs to be described more accurately. Imagine organization A with the patent history of code1, code2 and code3 and organization B with the patent history of code3, code4, and code 5. They create a patent together with code1, code3 and code4. Is this patent a recombination in your definition? It introduces a new technology for both firms that they have not used or combined previously by their own. Is it differentiated in any way from the situation when code1, code2, code3, code4 and code5 are all used in the collaboration-based patent?

9. I do not think the distance calculation formula is necessary in the main text. Please consider to move it to the SI/appendix.

10. The dyad level independent variables are averages of the respective values. How would the results change in case summarized values would be used alongside controls for absolute differences in regressions?

11. Due to the bipartite-to-unipartite network projection of patent-IPC code combinations, the knowledge network of companies will be highly clustered. Could you please show the distribution of the variable? I am suspicious that many companies have a knowledge network that does not show high clustering, but consist of several network components. Is this true? Do companies produce patents in unconnected areas that collaborations make connected? Could you please control for the number of knowledge network components?

12. A possibly better alternative for the decomposability (clustering) variable would be the ratio of clustering and clustering in a random network with similar size.

6. PLOS authors have the option to publish the peer review history of their article (what does this mean?). If published, this will include your full peer review and any attached files.

Reviewer #1: No

Reviewer #2: No

---

## [Author Response · Author response to Decision Letter 0]

3 Oct 2023

RESPONSE LETTER: PONE-D-22-33904

Dear editors and reviewers,

Thank you for giving us the opportunity to submit a revised draft of our manuscript titled “The influence of proximity and knowledge base on recombination innovation in R&D Collaboration” to Plos One. We appreciate the time and effort you and the reviewers have dedicated to providing your valuable feedback on our manuscript. We are grateful to the reviewers for their insightful comments on our paper. 

We have been able to incorporate changes to reflect most of the suggestions provided by the reviewers. We have taken your advice seriously and framed the manuscript. Below you can find a point-by-point response to your comments. Again, we thank you wholeheartedly for allowing us to resubmit a major revision of our study. 

Reviewer 1 comments：

In their paper, the authors investigate the effects of proximity (geographical and technical) on the recombination innovation at the dyad level between companies. Additionally, the authors test how this effect is mediated by an organization's knowledge base structure. Thus, based on theories from the R&D field, the authors propose a theoretical framework that explains the interactive effect of proximity and knowledge network structure on companies' collaborative recombination innovation potential. They derive six hypotheses that test for the direct and interactional effects of the aforementioned variables. For their empirical evaluation, the authors utilize longitudinal patent data from 150 organizations in the global nanotechnology industry. After conducting their statistical analyses, the authors conclude that technological proximity exerts a negative effect on the recombination innovation of companies while geographic proximity has an inverted U-shaped effect. Moreover, they were able to show that an organization's knowledge base decomposability plays a negative role in moderating the effect of technological proximity while playing a positive role in regulating the effect of geographic proximity. Degree centrality on the other side positively moderates the effect of both technological and geographic proximity.

In this review, I will highlight the research’s strengths and weaknesses and outline the challenges and opportunities presented in the manuscript.

I enjoyed reading the manuscript and appreciate the authors’ work. They succeed in tackling an interesting subject by developing a comprehensive theoretical framework and conducting thorough empirical analyses. By discussing different potential effects of the indicators under scrutiny, the authors derive six plausible hypotheses and test them convincingly in their empirical analyses. Additionally, the included figures and tables are clearly arranged and help the reader to follow their conceptual as well as empirical procedure. Overall, the authors use reasonable language, which makes it possible to follow their presented ideas most of the time.

While the paper succeeds in delivering its main messages, my review reveals potential issues that might diminish the contribution of the presented manuscript:

1. No direct research question

In their introduction, the authors mention that they aim to analyze "how knowledge interdependence at the micro level and partners’ knowledge origin interact leading to the change in state of collaborative recombination innovation." Although they mention their research goal, it would be helpful if they could include precise research questions which would make it easier for the reader to understand the overall goal of the paper.

Thank you for your constructive and useful review. The motivation of the study is unclear in the previous version of the paper. In the revised version, we rewrote the Introduction part to illustrate the research question of the study more clearly. 

We put forward the practical question “there are a large number of cases of dissolution and failure of inter-organizational collaborations. The high failure rate of inter-organizational collaborations restricts the development of firms’ recombination innovation.” 

Then, we also put forward the research question “How partner selection affect firms’ collaborative recombination innovation?” and also briefly introduce the main analytical lens of this paper as “this paper attempts to explore the characteristics of collaborative partners that influence firms’ collaborative recombination innovation based on the proximity perspective as well as the characteristics of organizations’ knowledge base”.

2. Introduction of theories

When introducing their concept of proximity, the authors mention the analytical framework of Boschma with its five dimensions of proximity. Yet, they never explain those five dimensions in detail. To help the reader understand their theoretical framework more thoroughly, the authors should introduce their utilized theories in more detail.

Thank you for your constructive and useful review. As you suggested, we added explanations about the Boschma’s analytical framework of five dimensions of proximity on Page 6.

3. Operationalization of variables

It has to be positively emphasized that the authors operationalize all mentioned variables by including a measurement description thereof. Yet, in several cases (e.g. technological proximity or control variables like structural hole) the authors do not explain how the measurement relates to the theoretical concept. It would be helpful if the authors could highlight the link between concept and measurement more thoroughly. When mentioning their formula to calculate the spatial distance, the authors do not explain the meaning of the number "6357." Please explain this formula in more detail. Additionally, if the authors establish a link between concept and measurement (e.g. knowledge base decomposability), the paper could be enhanced if the authors could provide more information regarding the theoretical background (e.g. what is the four-digital level class of IPC?).

Thank you for raising this important point. As you suggested, we added explanations about the independent and control variables in detail on Page 20, 22, and 23. In this way, we explain the link between concept and measurement more thoroughly. 

As suggested by the reviewer, we moved the description on the measurement of geographic distance to the Appendix. We explained the meaning of 6357 as “6357 is the polar radius” on Page 39. 

We gave theoretical background of technological classes on Page 19 and also explained the meaning of the four-digit level IPC class on Page 22. We also give the theoretical background of other main variables such as degree centrality and recombination innovation. 

4. Creation of dataset

When curating their dataset, the authors mention that they exclude patents that are filed to individual inventors. Yet, they do not support this decision with any arguments. Likewise, the authors state that they aim to incorporate an organization's history by only including organizations that had at least one patent application before the observation year. In this case, the authors provide a justification for their decision, yet it would be helpful if they could elaborate on why it is important to take an organization's history into account and why they decided for this specific period.

Thank you for raising this important point. We explain why we exclude individual inventor’s patents by stating that “as our analysis is conducted on organizational level and we focus on organizations and their granted patents, thus patents filed to individual inventor are excluded from the dataset” on Page 18. 

We elaborate on why it is important to take an organization’s history into account by stating “It is because several measurements in our analysis requires the organization’s patenting history. For example, we rely on organizations’ previous patents to obtain their technology portfolio and calculate technological proximity. We also rely on organizations’ previous patents to calculate accumulated patent counts and obtain the measure of past performance. We started our analysis from the year of 2000 as from this year on, nanotechnology patents increased dramatically. We ended our analysis in 2018 as our patent data is only updated till 2018.”

5. Statistical analysis

Although the authors mention that they test for potential multicollinearity with the variance inflation factor (VIF), they do not control for any other assumptions necessary to conduct a logistic regression (e.g. independence of observations, no outliers, linear relationship between explanatory variables and the logit of the response variable). To obtain sound results, the authors should check for these assumptions since they could affect their empirical results and thus their theoretical contribution. Additionally, it would be helpful if the authors could include the R-squared value and the F1 score. Otherwise, it is hard to determine how suitable the model is for their data (how much variance in the dependent variable can be explained by the model?). Currently, the authors perform aggregated analyses with their data ranging from 2000-2018. To receive a more dynamic understanding of the relationship between the different variables, it would be interesting to perform longitudinal analyses since the authors work with time-series data.

We are grateful for your advice. In the revised version, as you suggested, we re-run the regression analysis using the random-effect panel Logit model to have a more dynamic understanding of the relation. The results are displayed in Table 5. We also updated the Descriptive analysis and correlations of variables accordingly in Table 3 and 4. 

Before the Logit analysis, we perform several tests including the Hausman test, the Durbin-Watson test, the calculation of Cook's Distance of the variables, and OLS regression. These tests can ensure the independence of observations, no outliers in the variables, and the linear relationship between explanatory variables and the logit of the response variable. The description is on Page 26.

We included the Wald chi2, Prob>chi2, and Log likelihood of the regression analysis. Prob>chi2 can reflect how much variance in the dependent variable can be explained by the model.

6. Exemplary issues regarding language and format:

Grammar and wording:

- In a lot of cases, the authors forgot to mention the respective article in front of a noun (e.g. "Then, in the purpose of opening the “black box” of relation between proximity and recombination innovation, we viewed knowledge base as a structural factor during technological adaptation process among organizations."); please check the entire manuscript.

- p.1: "While to date, our comprehension of how partnering organizations access a broader knowledge pool, uncover novel perspectives, and spot recombination opportunities on the technological landscape is limited [6, 7]."; the word "while" needs to be deleted, otherwise one would expect a counter-argument at the end of the sentence.

- p.2: "Notably, existing studies has highlighted the importance of organizations’ external R&D collaboration as a critical mechanism to recombine novel knowledge from partners, the process of which would spur more recombination innovation."; the "has" needs to be changed into "have."

- p.2: I do not understand the following sentence: "Knowledge network represents the coupling logic of organizations and guilds organizations knowledge searching process."; please rewrite this sentence.

- p.2: "Previous studies puts forward that knowledge base is a network of knowledge element in essence [6, 19, 21, 22] that has great implications for formation of collaborations [20, 23], collaborative effectiveness [18, 20], and learning of partners’ knowledge [9]."; "puts" needs to be transformed into "put", "element" needs to be the plural form, "has" needs to be changed to "have", and there is an article missing before formation.

- p.3: "We detailly concentrate on two marked aspects of knowledge network which are knowledge base decomposability i.e., the extent to which knowledge elements are connected with each other forming clustered structure, and degree centrality of knowledge elements i.e., the extent to which knowledge elements are connected to other knowledge elements within and beyond organizational boundary."; to my knowledge, the word "detailly" does not exist, please revise. Additionally, "structure" should be transformed into the plural form.

- p.7: "While, too much similarities in knowledge base would hurt the alternative knowledge that can be considered limiting the recombination scope [47]."; the authors should delete the "while" and "much" needs to be changed to "many."

- p.9: "Therefore, locating in a vicinity of collaborating partners always means a high level of relationship quality, promotes the mutual understanding, and supports creativity and information exchange."; "locating in a vicinity" is hard to understand, simply rewrite it to e.g., "being spatially close to collaborating partners."

- p.20: "It means that large organizations and large research teams do not well perform in the realm of recombination innovation."; please change "well perform" to "perform well."

We are grateful for your advice. In the revised version, we revised the paper according to your suggestions and corrected the grammar and wording in the paper.

References

- p.26: "However, previous studies stated that joint patenting can be seen as the result of enduring R&D collaboration between organizations."; please provide some sources when mentioning "previous studies."

We added two references [66, 67] in this section. 

[66] Lin C, Wu Y-J, Chang C, et al. The alliance innovation performance of R&D alliances—the absorptive capacity perspective. Technovation 2012; 32: 282–292.

[67] Fritsch M, Titze M, Piontek M. Identifying cooperation for innovation―a comparison of data sources. Industry and Innovation 2020; 27: 630–659.

Format

- It seems like formulas were included with a screenshot and are therefore of low-resolution quality. Please check and insert them directly in the document. If possible, also increase the resolution of the inserted figures.

Thanks for your suggestions. We re-wrote the formulas in the paper using Word Formula. 

7. Conclusion

In summary, I can state that the authors pursue an interesting line of research that highlights the importance and different effects of proximity on the recombination innovation of companies on a dyad level. Yet, there are some issues, particularly regarding their operationalization of variables and their statistical analyses that could be improved. I hope my suggestions are helpful and wish them success as they continue their research.

Thank you for your continued support and guidance!

Reviewer 2 comments：

The paper illustrates the connection between proximity of firms and the creation of recombination-based patents at the dyad-level. Proximity of firms is considered through geographic proximity, technological proximity and the similarity of company-level knowledge bases.

I find the basic idea behind the paper interesting, however, I have a list of comments and suggestions for improvement below:

1. The first sentence of the abstract is very hard to follow. I would suggest to please the readers and catch their attention with an easy-to-understand beginning.

Thank you so much for these comments. We rewrote the first sentence of the abstract. 

2. The applied methodology is not mentioned/hinted in the abstract at all.

In the abstract, we added the methodology used in the paper.

3. The Proximity, knowledge base and recombination innovation section of the paper does not refer to the growing literature about atypical combinations and teamwork. I would suggest to build on this stream of literature to strengthen the motivation of the paper. Why are recombination innovation important in general?

Uzzi et al. (2013) Atypical combinations and scientific impact. Science 342 (6157), 468-472.

Thank you for raising the point. We included this reference in the revised version of the paper. 

In the introduction section on Page 1 we added:

“Studies also show that the atypical combination of conventional knowledge spanning a broader range of technological domains often produces inventions with unusual high impacts [3].” 

4. In general, the paper would benefit from descriptive figures that illustrate the sample and the key concepts. For example, network metrics are applied, so an illustrative example network (or two, since the focus is on dyads) could make the paper more appealing and easier to follow. It would be nice to see the distribution of the number of patents and the number of technology classes by companies in the final sample.

Thank you for pointing us to this limitation. In the revised version of the paper, we plotted the example knowledge network (illustrating both high and low level of knowledge base decomposability and degree centrality) in Figure 2 and 3 on Page 26.

We analysed the distribution of the number of patents and the number of technology classes by companies in the final sample. We show some descriptive indexes in Table 1 on Page 25. 

6. Additionally, illustration on how balanced are dyads in terms of patents by company and IPC classes filed by company would be nice to see. Is it the case that large, productive and often specialized companies tend to introduce new combinations in cooperation with smaller, more innovative, but less experienced firms? In such a situation, a small firm with a single patent would have a highly clustered knowledge network by definition. Clear descriptive on this regard would help.

We show some descriptive indexes about the distribution of the number of patents and the number of technology classes at the level of collaborative dyads in Table 2 on Page 25. We analysed the indexes on Page 25 as follows:

“Table 2 presents the distribution of the number of patents and the number of technology classes of collaborative dyad organizations. As shown, in the subgroup of co-patenting organizations that does not produce recombination innovation, the dyad organizations have the average patent number of 34 and 26 (Technological classes: 14 and 10). Comparably, in the subgroup of co-patenting organizations that produces recombination innovation, the dyad organizations have the average patent number of 13 and 13 (Technological classes: 7 and 20). It shows that, on average, specialized organizations tend to introduce new combinations in cooperation with other similar-sized organizations with diverse sets of technologies.”

7. The dependent variable (recombination innovation) only considers combinations that are new to the two companies in the focal dyad. Since patents represent highly codified innovation, firms can put resources together to more effectively copy an already existing solution (or moved towards a technological solution). Could you illustrate how often the observed dyads introduce new combinations that are novel for the entire sample of companies? Focusing only on absolute novel combinations might lead to a too small sample size in your context (?), however, discussing your results in relation to other studies highlighting the role of geography for the first combination of technology classes on patents might improve the paper.

Juhász et al. (2020) Explaining the dynamics of relatedness: The role of co-location and complexity. Papers in Regional Science 100 (1), 3-21.

Thank you for your suggestions. We calculated recombination innovation that is new to the whole sample. This measurement of recombination innovation can capture inventions that contain a higher level of novelty. Our measurement shows that the probability that dyads introduce new combinations that are novel for the entire sample of companies is very low. Among the whole sample of 450 co-patents, only 11 patents that have knowledge combinations that are novel for the entire sample of companies are observed. It would lead to too low probability to conduct regression analysis. 

As you suggested, we include this reference in our paper and discuss the implication of geography on the first combination of technological classes on Page 31.

8. The dependent variable (recombination innovation) needs to be described more accurately. Imagine organization A with the patent history of code1, code2 and code3 and organization B with the patent history of code3, code4, and code 5. They create a patent together with code1, code3 and code4. Is this patent a recombination in your definition? It introduces a new technology for both firms that they have not used or combined previously by their own. Is it differentiated in any way from the situation when code1, code2, code3, code4 and code5 are all used in the collaboration-based patent? 

Our definition of recombination innovation includes the case that a new IPC is introduced in the co-patents. We explain the specific measurement of recombination innovation in detail on Page 19 and 20 as follows:

“A combination of IPC class is considered new if no patents of the two organizations in the previous years had the same combination of IPC classes. For example, if organization A possess IPC class X and Z. Organization B possess IPC class Y and K. If organization A and B collaborate and create patent 1 with IPC class X, Y and patent 2 with IPC class M, then these two patents are all counted as recombination innovation jointly developed by A and B. It is because patent 1 contains the combination of X and Y that has not appeared in the previous patents of both organization A and B. Patent 2 introduces a new technology M for both firms that they have not used or combined previously by their own. In this way, our measurement of recombination innovation not only includes co-patents that contain a new IPC classification that has not appeared in the previous patents of collaborating organizations, but also the new combinations of existing IPCs.”

9. I do not think the distance calculation formula is necessary in the main text. Please consider to move it to the SI/appendix.

Thanks for your suggestions. We moved the distance calculation formula to Appendix 1. 

10. The dyad level independent variables are averages of the respective values. How would the results change in case summarized values would be used alongside controls for absolute differences in regressions?

Thanks for your suggestions. We added a robustness check section on Page 29. In this analysis, we utilized the alternative measurements of control variables including the sum and difference of values of collaborating organizations’ past patent performance, knowledge variety, age, and the usage of internet. The results of the robustness check analysis are displayed in Table 6. The results are similar to our main results. 

11. Due to the bipartite-to-unipartite network projection of patent-IPC code combinations, the knowledge network of companies will be highly clustered. Could you please show the distribution of the variable? I am suspicious that many companies have a knowledge network that does not show high clustering, but consist of several network components. Is this true? Do companies produce patents in unconnected areas that collaborations make connected? Could you please control for the number of knowledge network components?

Thanks for pointing out this concern. The mean value of clustering is 0.82 with a minimum value of 0.36 and a maximum value of 1. The measurement shows that most of the organizations have a clustering around 0.3-1. The average clustering is high. There are cases that organizations have very 1-2 patents thus having a clustering of 1. We didn’t exclude these cases. But we found that the exclusion of these cases did not affect our results. 

As you suggested, in the revised version of the paper, we controlled for the number of knowledge components in an organization’s knowledge network using the variable “Knowledge element”. 

12. A possibly better alternative for the decomposability (clustering) variable would be the ratio of clustering and clustering in a random network with similar size.

Thanks for your suggestions. We added a robustness check section on Page 29. In this analysis, we utilized the alternative measurements of knowledge base decomposability using the ratio of the clustering coefficient of organizations’ knowledge network and clustering in a random network with the same number of vertices and edges. The results do not support Hypothesis 3. But the sign of the regression coefficient is the same as the regression results in the paper (-7.98). The results support Hypothesis 4.

---

## [Decision Letter · Decision Letter 1]

24 Oct 2023

PONE-D-22-33904R1The influence of proximity and knowledge base on recombination innovation in R&D CollaborationPLOS ONE

Dear Dr. Nan,

Thank you for submitting your manuscript to PLOS ONE. After careful consideration, we feel that it has merit but does not fully meet PLOS ONE’s publication criteria as it currently stands. Therefore, we invite you to submit a revised version of the manuscript that addresses the points raised during the review process.

We look forward to receiving your revised manuscript.

Kind regards,

Claudia Noemi González Brambila, Ph.D.

Academic Editor

PLOS ONE

Journal Requirements:

Reviewers' comments:

Reviewer's Responses to Questions

**Comments to the Author**

1. If the authors have adequately addressed your comments raised in a previous round of review and you feel that this manuscript is now acceptable for publication, you may indicate that here to bypass the “Comments to the Author” section, enter your conflict of interest statement in the “Confidential to Editor” section, and submit your "Accept" recommendation.

Reviewer #1: (No Response)

Reviewer #2: All comments have been addressed

2. Is the manuscript technically sound, and do the data support the conclusions?

Reviewer #1: Yes

Reviewer #2: Yes

3. Has the statistical analysis been performed appropriately and rigorously? 

Reviewer #1: Yes

Reviewer #2: Yes

4. Have the authors made all data underlying the findings in their manuscript fully available?

Reviewer #1: Yes

Reviewer #2: Yes

5. Is the manuscript presented in an intelligible fashion and written in standard English?

Reviewer #1: Yes

Reviewer #2: Yes

6. Review Comments to the Author

Reviewer #1: I thank the authors for this revision. While I am overall satisfied with your edits and the current version of the paper, I will point out some minor issues that could be addressed more concisely. In the following, I will provide additional thoughts that could improve the quality of the paper.

1. Unclear research gap

In their introduction, the authors write, "However, despite the recognition of the proximity effect on collaborative innovation, current research fails to throw light upon two essential aspects." It remains unclear what concrete two aspects the authors refer to as they continue the next paragraph with a new topic. The authors could strengthen their argument and identified research gap by naming the two essential aspects directly (which should naturally follow their previous argumentation).

2. More supporting arguments for deciding on the two aspects of the knowledge network

The authors state in their introduction that they primarily focus on knowledge base decomposability and degree centrality of knowledge elements when evaluating the interaction effects of the knowledge graph on the relationship between technological and geographical proximity on recombination innovation. Unfortunately, the authors do not include any arguments supporting that decision. Why did they decide on these aspects of the knowledge graph and excluded others? The quality of the paper could be improved if the authors could add more arguments for their decision.

3. Potential interaction between technological and geographic proximity

On page 11, the authors write, "The reason lies in that when collaborating agents are located too close, similar technical backgrounds and capabilities result in the draining of knowledge variety and standardization of know-how." I agree that geographical proximity potentially leads to more similar technical backgrounds and capabilities. Yet, this similarity likely impacts the technological similarity of organizations. Thus, I am wondering if both independent variables might be intercorrelated on a conceptual level (as the authors tested for multicollinearity and did not find worrisome results on an empirical level), affecting recombination innovation. Hence, it would be helpful if the authors could include more arguments that sustain the fact that both concepts are independent of each other and, therefore, essential to determine recombination innovation.

4. Operationalization of knowledge base decomposability

The authors explain that they measure knowledge base decomposability with the "percentage of closed triads in the sum of closed and open triads in the network according to the formula below." As this formula represents the global/overall clustering coefficient, which the authors also mention later on, I suggest using this naming from the beginning, making clear that "knowledge base decomposability" is a concept that is operationalized through the global/overall clustering coefficient. For more information, please see:

Jackson, M. O. (2008). Social and economic networks (Vol. 3). Princeton: Princeton University Press.

5. Clarifying degree centrality

The authors measure the degree centrality of knowledge elements by summing up the degree centrality of all knowledge elements. I assume the authors correctly apply the normalized degree centrality as they compare values across graphs. It would be helpful if the authors could mention this information within the manuscript.

6. Statistical robustness test

I acknowledge the authors' effort of having included several statistical robustness tests for their statistical model. I want to add some minor remarks that could potentially improve the quality of this section. First, as not all readers are familiar with the statistical test applied, please briefly describe each test's purpose (e.g., why did you apply the Hausmann test?). Second, you mention on p. 26 that you calculated the VIF without providing any results. On p. 27, you refer again to the VIF and mention the respective results. Please resolve this redundancy and immediately state the results of the VIF and the potential implications (which are none in your case). Third, I am wondering how you used an OLS to check for a linear relationship within the model. Did you do a visual analysis? It would be great if you could elaborate on your approach in more detail. Fourth, you write that "... the past performance of the organizations is negatively related to the recombination innovation in collaboration." I am not sure what you mean by "performance." It would be great if you could state more explicitly what variables relate to performance.

7. Results of regression with alternative measures

To test the robustness of their model, the authors conduct an additional regression with alternative measures. I think this additional robustness check is a valuable addition to the paper and improves its quality. The authors state that the additional regression yields similar results compared to their base model. To improve the quality of the paper even further, it would be great if the authors could elaborate in more detail if all of the main effects stay the same and, if not, provide some potential explanations for this finding.

8. Minor issues

- Gramma:

p. 0: "We validated the theoretical hypotheses using the Logit regression models based on the longitudinal data of 150 organizations in the global nanotechnology industry."; please delete "the" as it is not necessary in this sentence.

p. 0: "By contrast, the degree centrality of the knowledge element positively moderates the effect of both technological and geographic proximity."; please exchange "By" with "In."

p. 0: "In the last years, recombination innovation has been assuming increasing importance since the miniaturization ... "; please exchange "been assuming" with "gained."

p. 3: "... lays the foundation for an organizations ability to recombine partners’ knowledge... "; please write "organization's."

p. 6: "With the notion of cognitive proximity or technological proximity, actors share the same knowledge base and thus able to..."; please include "are" between "thus" and "able."

p. 6: "on the other hand, social proximity..."; please capitalize "On."

p. 18: "Then, as our analysis is conducted on organizational level and we focus on organizations and their granted patents, thus patents filed to the individual inventors are excluded from the dataset. Our initial set of data consists of a total of 11000 patents."; please delete the "thus" in this sentence.

p.20: "For example, if organization A possess IPC class X and Z. Organization B possess IPC class Y and K."; please combine these two sentences into one and rewrite "possesses."

p.24: "In addition, the level of utilization of theInternet might influence the effect of geographic proximity on collaborative innovation."; please include a space between "the" and "Internet."

p. 24: "Different types of organizations may different innovative patterns ..."; please include "have" between "may" and "different."

- References

Some of the references contain DOIs, and others do not. Please check and use a consistent citation style.

9. Conclusion

In summary, I am satisfied with the authors' edits and the current version of the paper. Yet, I raised a few issues that could be addressed to improve the quality of the paper. I hope these suggestions are helpful for the authors and wish them success as they continue their research.

Reviewer #2: Thanks for the thorough revision! In my eyes, the paper improved a lot, but I would suggest further - mainly minor - changes.

Suggestions for improvement:

1) “In addition, in the existing literature, proximity and collaborative innovation are directly linked in theorization, while the underlying transmission mechanism is, to a large extent, black-boxed [13, 14]. “

I would use this sentence to illustrate a possible improvement of this work in terms of positioning. Studies have already used patent data to investigate the role of proximity in innovation related collaboration (co-patenting, co-publication, ...). However, this is often done at the regional level (e.g. Tóth et al. 2021) or at the individual level (e.g. Ter Wal et al. 2013).

I think that your study provides new insights into what drives a firm-to-firm (organization-to-organization) relationship to become an innovation. I think it would help the study if you could articulate this better around the abstract/introduction/discussion. In particular, I would highlight your focus on firm-firm relationships and you are interested in the characteristics of organizations that lead to innovation/patenting.

References:

Tóth, G., Juhász, S., Elekes, Z. And Lengyel, B. (2021) Repeated collaboration of inventors across European regions. European Planning Studies, 29 (12), 2252-2272.

Ter Wal, A. L. J. (2014) The dynamics of the inventor network in German biotechnology: geographic proximity versus triadic closure. Journal of Economic Geography, 14 (3), 589-320.

2) In economic geography and related literature, "knowledge networks" are seen as networks of (formal/informal) knowledge exchange, and what you call in your paper a knowledge network is more properly called a "knowledge space".

With respect to knowledge networks, the impact of cognitive/technological proximity on the formation and maintenance of collaborations is well documented.

However, the application of the idea of "knowledge space" at the firm/organisational level and linking it to the outcomes of cooperation is (to my knowledge) new in the economic geography literature. I would suggest that you make your contribution clear and possibly position your setting relative to their previous findings.

Useful references for knowledge network evolution:

Balland, P. A., Belso-Martinez, J. A. and Morrison, A. (2016) The Dynamics of Technical and Business Knowledge Networks in Industrial Clusters: Embeddedness, Status, or Proximity? Economic Geography, 92 (1), 35-60.

Juhász, S. And Lengyel, B. (2018) Creation and persistence of ties in cluster knowledge networks. Journal of Economic Geography, 18 (6), 1203-1226.

Usefull references for knowledge space:

Balland, P. A., Rigby, D. and Boschma, R. (2015) The technological resilience of US cities. Cambridge Journal of Regions, Economy and Society, 8, 167-184.

Hidalgo, C. A. (2021) Economic complexity theory and applications. Nature Reviews Physics, 3 (2), 92-113.

3) Following the example of recombination innovation on page 19. I argue that patent 2 with the new M technology is not a recombination innovation. Neither of the two companies have used this technology before, so they are not "recombining" their knowledge, but collaborating to learn something that is completely new to both of them.

How would your results change if you would remove cases of such recombination innovation, where we actually observe a "new to both" (and in some cases "only new codes to both") combinations?

7. PLOS authors have the option to publish the peer review history of their article (what does this mean?). If published, this will include your full peer review and any attached files.

Reviewer #1: No

Reviewer #2: No

---

## [Author Response · Author response to Decision Letter 1]

22 Dec 2023

RESPONSE LETTER: PONE-D-22-33904R1

Dear editors and reviewers,

Thank you for giving us the opportunity to submit a revised draft of our manuscript titled “The influence of proximity and knowledge base on recombination innovation in R&D Collaboration” to Plos One. We appreciate the time and effort you and the reviewers have dedicated to providing your valuable feedback on our manuscript. We are grateful to the reviewers for their insightful comments on our paper. 

We have been able to incorporate changes to reflect most of the suggestions provided by the reviewers. We have taken your advice seriously and framed the manuscript. Below you can find a point-by-point response to your comments. Again, we thank you wholeheartedly for allowing us to resubmit a minor revision of our study. 

Reviewer 1 comments：

1. Unclear research gap

In their introduction, the authors write, "However, despite the recognition of the proximity effect on collaborative innovation, current research fails to throw light upon two essential aspects." It remains unclear what concrete two aspects the authors refer to as they continue the next paragraph with a new topic. The authors could strengthen their argument and identified research gap by naming the two essential aspects directly (which should naturally follow their previous argumentation).

Thank you for your constructive and useful review. We added a paragraph to briefly discuss (also considering that the length of the introduction cannot be too long) and point out the main two theoretical issues unsolved in the existing studies as follows on Page 2-3.

“First, in the existing literature, proximity and collaborative innovation are directly linked in theorization, while the underlying transmission mechanism is, to a large extent, black-boxed [13, 14]. In the economic geography literature, studies have investigated various proximity dimensions, and their effects on the formation and maintenance of collaborations in the context of inter-regional [15] and inventor collaboration [16]. In addition, the effect of proximity on the knowledge sharing and production during inter-organizational alliance is well documented [17]. While in the theorization of the effects of proximity during organization-to-organization collaborations, the role of underlying organizational characteristics remains less salient. Research has rarely examined how proximity and organizational characteristics jointly spur innovation at the collaborative dyad level. Second, existing literature highlights that studies should focus on knowledge space within organizational boundaries when examining inter-organizational collaboration [18]. While, examining the current economic geography literature, few studies have applied the concept of "knowledge space" at the organizational level and link it to the outcomes of cooperation.”

2.More supporting arguments for deciding on the two aspects of the knowledge network

The authors state in their introduction that they primarily focus on knowledge base decomposability and degree centrality of knowledge elements when evaluating the interaction effects of the knowledge graph on the relationship between technological and geographical proximity on recombination innovation. Unfortunately, the authors do not include any arguments supporting that decision. Why did they decide on these aspects of the knowledge graph and excluded others? The quality of the paper could be improved if the authors could add more arguments for their decision.

Thank you for your constructive and useful review. As you suggested, we added explanations about why we choose these two types of knowledge base characteristics on Page 4. Mainly, we consider that these two types of knowledge base originated from an organization’s two distinct learning and innovative strategies. 

“These two dimensions of knowledge base characteristics of an organization can represent two different innovation strategies. Knowledge base decomposability usually originates from an organization’s expertise in specialized fields of knowledge [19, 21]. These organizations often rely on their deep utilization of domain knowledge to maintain existing advantages. The degree centrality of knowledge elements, on the other hand, usually originates from the broader connectedness of technologies within the organization [21, 23]. These organizations tend to rely more on the combinatorial potential of knowledge as well as external knowledge acquisition to exploit the internal technologies fully.”

3. Potential interaction between technological and geographic proximity

On page 11, the authors write, "The reason lies in that when collaborating agents are located too close, similar technical backgrounds and capabilities result in the draining of knowledge variety and standardization of know-how." I agree that geographical proximity potentially leads to more similar technical backgrounds and capabilities. Yet, this similarity likely impacts the technological similarity of organizations. Thus, I am wondering if both independent variables might be intercorrelated on a conceptual level (as the authors tested for multicollinearity and did not find worrisome results on an empirical level), affecting recombination innovation. Hence, it would be helpful if the authors could include more arguments that sustain the fact that both concepts are independent of each other and, therefore, essential to determine recombination innovation.

Thank you for raising this important point. In the previous studies on proximity, geographical proximity and technological/cognitive proximity are often independently studied. As you suggested, we added explanations about this issue as follows on Page 10. 

“Geographic proximity reflects a different dimension of relational characteristic among partnering organizations compared to technological proximity. Technological proximity represents the overlap of technological background and expertise of the organizations, while geographic proximity is the spatial distance between the collaborating organizations. Spatial propinquity encourages frequent face-to-face interactions, thus stimulating the emergence of interpersonal networks across the boundaries of partner organization. This, in turn, increases the opportunities for face-to-face communication of the inventors, facilitating the development of relational trust. Therefore, partnering organizations develop, learn, and adjust over time the idiosyncratic languages needed for the sharing of ‘fine-grained information’ and tacit knowledge. Although organizations located in the closed vicinity of each other also draw on a same set of regional knowledge and tend to share similar values, norms, and “thinking models”, it does not necessarily mean that these organizations have a larger technological proximity. Therefore, geographical proximity and technological proximity are independent from each other and are both essential to determine recombination innovation at the collaborative dyad level. ”

4. Operationalization of knowledge base decomposability

The authors explain that they measure knowledge base decomposability with the "percentage of closed triads in the sum of closed and open triads in the network according to the formula below." As this formula represents the global/overall clustering coefficient, which the authors also mention later on, I suggest using this naming from the beginning, making clear that "knowledge base decomposability" is a concept that is operationalized through the global/overall clustering coefficient. For more information, please see:

Jackson, M. O. (2008). Social and economic networks (Vol. 3). Princeton: Princeton University Press.

Thank you for raising this important point. As you suggested we clarify that the knowledge base decomposability is obtained using the global clustering coefficient of the network on Page 21.

5. Clarifying degree centrality

The authors measure the degree centrality of knowledge elements by summing up the degree centrality of all knowledge elements. I assume the authors correctly apply the normalized degree centrality as they compare values across graphs. It would be helpful if the authors could mention this information within the manuscript.

We are grateful for your advice. In the revised version, we clarified that we used the normalized degree centrality on Page 22. 

6. Statistical robustness test

I acknowledge the authors' effort of having included several statistical robustness tests for their statistical model. I want to add some minor remarks that could potentially improve the quality of this section. First, as not all readers are familiar with the statistical test applied, please briefly describe each test's purpose (e.g., why did you apply the Hausmann test?). Second, you mention on p. 26 that you calculated the VIF without providing any results. On p. 27, you refer again to the VIF and mention the respective results. Please resolve this redundancy and immediately state the results of the VIF and the potential implications (which are none in your case). Third, I am wondering how you used an OLS to check for a linear relationship within the model. Did you do a visual analysis? It would be great if you could elaborate on your approach in more detail. Fourth, you write that "... the past performance of the organizations is negatively related to the recombination innovation in collaboration." I am not sure what you mean by "performance." It would be great if you could state more explicitly what variables relate to performance.

Thanks for the suggestion. We state the results of VIF immediately after first stating it on Page 26. As you suggested, we further explain the purpose of Hausmann test, Durbin-Watson test, and Cook’s distant respectively on Page 26. We conduct OLS test before Logit regression to see whether there is leaner relations between the explanatory variables and the logit of the response variable. We use the scatter plot and linear fitting plot to verify the linear relation. The results show that there is a significant linear relation between explanatory variables and the logit of the response variable. We clarify this on Page 27. 

The past performance is a control variable in our study. It is measured using the number of accumulated patents of the organizations. We explain it clearly on Page 26. 

7. Results of regression with alternative measures

To test the robustness of their model, the authors conduct an additional regression with alternative measures. I think this additional robustness check is a valuable addition to the paper and improves its quality. The authors state that the additional regression yields similar results compared to their base model. To improve the quality of the paper even further, it would be great if the authors could elaborate in more detail if all of the main effects stay the same and, if not, provide some potential explanations for this finding.

Thanks for the suggestion. The results of the robustness check are same except for one main effect. We added a possible explanation of the result as follows on Page 28.

“The results show similar effects of the independent variables except that the moderation effect of decomposability on the relation between technological proximity and recombination innovation is not significant. The sign of the coefficient is same as the results. The possible explanation is that the measurement of global network clustering coefficient comparing to the random network usually has smaller variance as it takes into account network size and density. As a result, the moderation effect of decomposability is less obvious using this alternative measure.”

8. Minor issues

- Gramma:

p. 0: "We validated the theoretical hypotheses using the Logit regression models based on the longitudinal data of 150 organizations in the global nanotechnology industry."; please delete "the" as it is not necessary in this sentence.

p. 0: "By contrast, the degree centrality of the knowledge element positively moderates the effect of both technological and geographic proximity."; please exchange "By" with "In."

p. 0: "In the last years, recombination innovation has been assuming increasing importance since the miniaturization ... "; please exchange "been assuming" with "gained."

p. 3: "... lays the foundation for an organizations ability to recombine partners’ knowledge... "; please write "organization's."

p. 6: "With the notion of cognitive proximity or technological proximity, actors share the same knowledge base and thus able to..."; please include "are" between "thus" and "able."

p. 6: "on the other hand, social proximity..."; please capitalize "On."

p. 18: "Then, as our analysis is conducted on organizational level and we focus on organizations and their granted patents, thus patents filed to the individual inventors are excluded from the dataset. Our initial set of data consists of a total of 11000 patents."; please delete the "thus" in this sentence.

p.20: "For example, if organization A possess IPC class X and Z. Organization B possess IPC class Y and K."; please combine these two sentences into one and rewrite "possesses."

p.24: "In addition, the level of utilization of theInternet might influence the effect of geographic proximity on collaborative innovation."; please include a space between "the" and "Internet."

p. 24: "Different types of organizations may different innovative patterns ..."; please include "have" between "may" and "different."

We are grateful for your advice. In the revised version, we revised the paper according to your suggestions and corrected the grammar and wording in the paper.

- - References

Some of the references contain DOIs, and others do not. Please check and use a consistent citation style.

We are grateful for your advice. In the revised version, we revised the reference style according to your suggestions.

9. Conclusion

In summary, I am satisfied with the authors' edits and the current version of the paper. Yet, I raised a few issues that could be addressed to improve the quality of the paper. I hope these suggestions are helpful for the authors and wish them success as they continue their research.

Thank you for your continued support and guidance!

Reviewer 2 comments：

1. “In addition, in the existing literature, proximity and collaborative innovation are directly linked in theorization, while the underlying transmission mechanism is, to a large extent, black-boxed [13, 14]. “

I would use this sentence to illustrate a possible improvement of this work in terms of positioning. Studies have already used patent data to investigate the role of proximity in innovation related collaboration (co-patenting, co-publication, ...). However, this is often done at the regional level (e.g. Tóth et al. 2021) or at the individual level (e.g. Ter Wal et al. 2013).

I think that your study provides new insights into what drives a firm-to-firm (organization-to-organization) relationship to become an innovation. I think it would help the study if you could articulate this better around the abstract/introduction/discussion. In particular, I would highlight your focus on firm-firm relationships and you are interested in the characteristics of organizations that lead to innovation/patenting.

References:

Tóth, G., Juhász, S., Elekes, Z. And Lengyel, B. (2021) Repeated collaboration of inventors across European regions. European Planning Studies, 29 (12), 2252-2272.

Ter Wal, A. L. J. (2014) The dynamics of the inventor network in German biotechnology: geographic proximity versus triadic closure. Journal of Economic Geography, 14 (3), 589-320.

Thank you so much for these comments. We rewrote the theoretical contributions of our study in the introduction on Page 2-3 as follows.

“First, in the existing literature, proximity and collaborative innovation are directly linked in theorization, while the underlying transmission mechanism is, to a large extent, black-boxed [13, 14]. In the economic geography literature, studies have investigated various proximity dimensions, and their effects on the formation and maintenance of collaborations in the context of inter-regional [15] and inventor collaboration [16]. In addition, the effect of proximity on the knowledge sharing and production during inter-organizational alliance is well documented [17]. While in the theorization of the effects of proximity during organization-to-organization collaborations, the role of underlying organizational characteristics remains less salient. Research has rarely examined how proximity and organizational characteristics jointly spur innovation at the collaborative dyad level. Second, existing literature highlights that studies should focus on knowledge space within organizational boundaries when examining inter-organizational collaboration [18]. While, examining the current economic geography literature, few studies have applied the concept of "knowledge space" at the organizational level and link it to the outcomes of cooperation.

In order to fill these gaps, we focus on organization-to-organization collaboration to further contribute to strengthening this body of work on the collaborative nature of innovation and the role of proximity in this process. ”

We also added more clarification about the theoretical contribution in the abstract on Page 0 and discussion part on Page 31. We highlighted that our contribution is mainly in the filed of economic geography and recombinant search literature to better position the paper.

2. In economic geography and related literature, "knowledge networks" are seen as networks of (formal/informal) knowledge exchange, and what you call in your paper a knowledge network is more properly called a "knowledge space".

With respect to knowledge networks, the impact of cognitive/technological proximity on the formation and maintenance of collaborations is well documented.

However, the application of the idea of "knowledge space" at the firm/organisational level and linking it to the outcomes of cooperation is (to my knowledge) new in the economic geography literature. I would suggest that you make your contribution clear and possibly position your setting relative to their previous findings.

Useful references for knowledge network evolution:

Balland, P. A., Belso-Martinez, J. A. and Morrison, A. (2016) The Dynamics of Technical and Business Knowledge Networks in Industrial Clusters: Embeddedness, Status, or Proximity? Economic Geography, 92 (1), 35-60.

Juhász, S. And Lengyel, B. (2018) Creation and persistence of ties in cluster knowledge networks. Journal of Economic Geography, 18 (6), 1203-1226.

Usefull references for knowledge space:

Balland, P. A., Rigby, D. and Boschma, R. (2015) The technological resilience of US cities. Cambridge Journal of Regions, Economy and Society, 8, 167-184.

Hidalgo, C. A. (2021) Economic complexity theory and applications. Nature Reviews Physics, 3 (2), 92-113.

Thanks for your suggestion. We replace the knowledge network with the “knowledge space” in the revised version of the paper. We clarify on Page 3 and Page 22 that the knowledge space is constructed using the interdependence network formed among different knowledge elements (IPC). We also illustrate the theoretical contribution of the paper regarding the conceptualization of “knowledge space” on Page 34 as follows.

“Shifting the conceptual lens from technological components to the interdependence or structure of knowledge elements, we apply the idea of "knowledge space" at the organizational level and link it to the outcomes of cooperation.”

3. Following the example of recombination innovation on page 19. I argue that patent 2 with the new M technology is not a recombination innovation. Neither of the two companies have used this technology before, so they are not "recombining" their knowledge, but collaborating to learn something that is completely new to both of them.

How would your results change if you would remove cases of such recombination innovation, where we actually observe a "new to both" (and in some cases "only new codes to both") combinations?

Thank you for your suggestions. For a robustness check, we remove such cases when collaborative organizations create a parent that contains an IPC code that is new to both of them. The cases are relatively small only accounting for 10% of the whole sample. We run the regression with the alternative measure of recombination innovation and obtain similar results. We add this in the Robustness check section on Page 28 as follows. 

“We also utilized an alternative measure of recombination innovation. When collaborative organizations create a patent that contains IPC codes that is new to both of them, they are collaborating to learn something that is completely new to them. We remove cases of such recombination innovation to obtain an alternative measure. For example, if organization A possesses IPC class X and Z and organization B possesses IPC class Y and K. If organization A and B collaborate and create patent 1 with IPC class X, Y and patent 2 with IPC class M, then patent 1 is considered recombination innovation while patent 2 is not. In the regression results with the alternative measure of recombination innovation all of the main effects stay the same.”

---

## [Decision Letter · Decision Letter 2]

30 Jan 2024

The influence of proximity and knowledge base on recombination innovation in R&D Collaboration

PONE-D-22-33904R2

Dear Dr. Nan,

We’re pleased to inform you that your manuscript has been judged scientifically suitable for publication and will be formally accepted for publication once it meets all outstanding technical requirements.

Kind regards,

Claudia Noemi González Brambila, Ph.D.

Academic Editor

PLOS ONE

Additional Editor Comments (optional):

Reviewers' comments:

Reviewer's Responses to Questions

**Comments to the Author**

1. If the authors have adequately addressed your comments raised in a previous round of review and you feel that this manuscript is now acceptable for publication, you may indicate that here to bypass the “Comments to the Author” section, enter your conflict of interest statement in the “Confidential to Editor” section, and submit your "Accept" recommendation.

Reviewer #2: All comments have been addressed

2. Is the manuscript technically sound, and do the data support the conclusions?

Reviewer #2: Yes

3. Has the statistical analysis been performed appropriately and rigorously? 

Reviewer #2: Yes

4. Have the authors made all data underlying the findings in their manuscript fully available?

Reviewer #2: Yes

5. Is the manuscript presented in an intelligible fashion and written in standard English?

Reviewer #2: Yes

6. Review Comments to the Author

Reviewer #2: (No Response)

7. PLOS authors have the option to publish the peer review history of their article (what does this mean?). If published, this will include your full peer review and any attached files.

Reviewer #2: No

---

## [Editor Report · Acceptance letter]

19 Feb 2024

PONE-D-22-33904R2 

PLOS ONE

Dear Dr. Nan, 

I'm pleased to inform you that your manuscript has been deemed suitable for publication in PLOS ONE. Congratulations! Your manuscript is now being handed over to our production team.

Kind regards, 

on behalf of

Dr. Claudia Noemi González Brambila 

Academic Editor

PLOS ONE